# Adaptive responses of animals to climate change are most likely insufficient

Viktoriia Radchuk (iD) et al.[#]

Biological responses to climate change have been widely documented across taxa and regions, but it remains unclear whether species are maintaining a good match between phenotype and environment, i.e. whether observed trait changes are adaptive. Here we reviewed 10,090 abstracts and extracted data from 71 studies reported in 58 relevant publications, to assess quantitatively whether phenotypic trait changes associated with climate change are adaptive in animals. A meta-analysis focussing on birds, the taxon best represented in our dataset, suggests that global warming has not systematically affected morphological traits, but has advanced phenological traits. We demonstrate that these advances are adaptive for some species, but imperfect as evidenced by the observed consistent selection for earlier timing. Application of a theoretical model indicates that the evolutionary load imposed by incomplete adaptive responses to ongoing climate change may already be threatening the persistence of species.

Climate change can reduce the viability of species and associated biodiversity loss can impact ecosystem functions and services[1–3]. Fitness losses (i.e. reductions in survival or reproductive rates) can be mitigated, however, if populations respond adaptively by undergoing morphological, physiological or behavioural changes that maintain an adequate match—or at least reduce the extent of mismatch—between phenotype and environment. Such adaptive phenotypic changes —which we call 'adaptive response' (to climate change)—come about via phenotypic plasticity, microevolution or a combination of both, and can occur in tandem with geographic range shifts[4–6]. Quantifying adaptive responses, or demonstrating their absence despite directional selection, is important in a biodiversity conservation context for predicting species' abundances or distributions[4,5] and for mitigating the effects of climate change on biodiversity by developing strategies tailored to species' ecologies[4–6].

Longitudinal studies of wild populations provide the opportunity to determine whether phenotypic changes are adaptive (e.g. refs. [7–9]). A phenotypic change qualifies as an adaptive response to climate change if three conditions are met: (1) a climatic factor changes over time, (2) this climatic factor affects a phenotypic trait of a species and (3) the corresponding trait change confers fitness benefits (Fig. 1)[10,11]. These conditions are usually assessed in isolation[11–14] (but see, for example, refs. [7,8]) and hence most studies can only speculate on whether adaptive responses have occurred. Here, we extracted data from many published studies to assess these three conditions in free-living animals and thus determine whether the observed phenotypic changes are adaptive.

Multiple studies report data satisfying the first two conditions. In particular, increases in temperatures across multiple locations during recent decades are well documented (i.e. global warming[15]). Similarly, the effects of climate change on several traits are well characterized. For example, the timing of biological events, such as reproduction or migration (hereafter 'phenological traits'), has generally advanced across multiple taxa and locations[13,16–18]. 'Morphological traits', such as body size or mass, have also responded to climate change, but show no general systematic pattern[8,14,19,20].

A substantial challenge to test the third condition is that the data must be collected over multiple generations in single populations. Existing datasets assembling data on either trait variation[21–23] or selection[24–26] across taxa, although valuable, are not well suited for testing whether phenotypic trait changes are adaptive, because these two types of datasets rarely overlap in terms of species, traits, study location and study period. Recently,

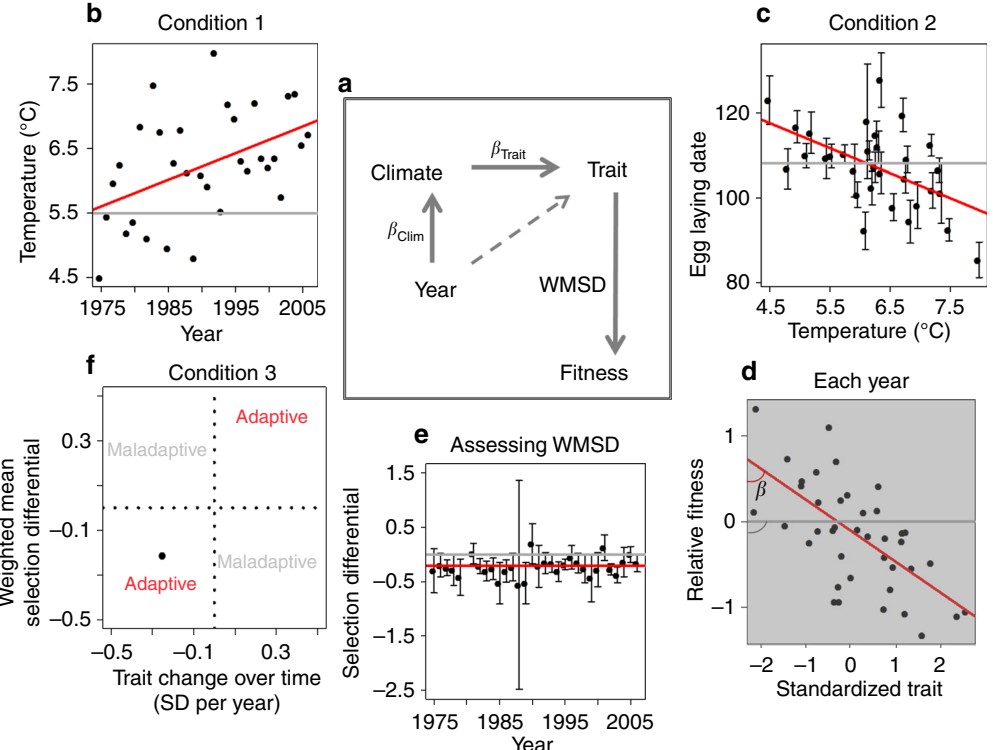

**Fig. 1** A framework for inferring phenotypic adaptive responses using three conditions. **a** General framework. Arrows indicate hypothesized causal relationships, with dashed arrow indicating that we accounted for the effects associated with years when assessing the effect of climate on traits. **b**–**f** demonstrate steps of the framework using as an example one study from our dataset—Wilson et al.[69]. **b** Condition 1 is assessed by $\beta_{Clim}$, the slope of a climatic variable on years, **c** Condition 2 is assessed by $\beta_{Trait}$, the slope of the mean population trait values on climate. **d** Interim step: assessing the linear selection differentials ($\beta$). Note that each dot here represents an individual measurement in the respective year and not a population mean; analyses of selection were not performed here but in original publications, except for a few studies, thus inset **d** is a conceptual depiction and not based on real data. **e** To assess condition 3, first the weighted mean annual selection differential (WMSD) is estimated. **f** Condition 3 is then assessed by checking whether selection occurs in the same direction as the trait change over time, calculated as the product of the slopes from conditions 1 and 2. Red lines and font in **b**–**f** illustrate the predictions from model fits. Grey lines and font illustrate the lack of effect in each condition. As an example, if temperature increased over years (as shown by the red line in **b**), phenology advanced (depicted by the red line in **c**) and WMSD was negative (as depicted by the red line in **e**), then fitness benefits are associated with phenological advancement, reflecting an adaptive response (point falls in quadrant 3 in **f**). Source data are provided as a Source Data file

Siepielski et al.[27] assembled a global dataset combining climatic factors and selection in species and showed that precipitation, rather than temperature, explained most of the variation in selection. However, neither their analysis nor a follow-up study[28] assessed whether phenotypic responses to climate (PRC) were adaptive because the assembled dataset does not contain data on trait changes.

We conducted a systematic literature search to assemble the necessary data to assess whether trait changes in response to climate change are adaptive across animal species worldwide. We mainly investigated the phenotypic responses of birds, because complete data on other taxa were scarce. We demonstrate that advancement of phenology is adaptive in some bird species, but this response is not universal. Further, modelling suggests that even bird species responding adaptively to climate change may adapt too slowly to be able to persist in the longer term.

## Results

**Systematic literature search.** Our literature search focused on studies that investigated how change in temperature, precipitation or both affects morphological or phenological traits in arachnids, insects, amphibians, reptiles, birds or mammals. To assess all three conditions necessary for inferring adaptive responses to climate change (Fig. 1), we selected publications reporting the following data from natural populations during at least 6 years: (1) annual values of a climatic variable, (2) annual mean values (+SE) of a phenotypic trait at the population level and (3) annual linear selection differentials measured on the trait(s). Annual linear selection differentials were measured as the slope of relative fitness on standardized trait values[29] (Methods, Fig. 1d) and reported for at least one of the three fitness components: adult survival, reproduction (measured as number of offspring) or recruitment (measured as number of offspring contributing to the population size the following year).

A search on Web of Knowledge (Methods) returned 10,090 publications, of which 58 were retained. These publications reported data on 4835 studies (representing 1413 non-aquatic species in 23 countries) that contained information on phenotypic responses to climate change. Out of these 4835 studies, a subset of 71 studies (representing 17 species in 13 countries) contained all the information required to assess whether responses were adaptive (including selection differentials, Methods). We stored information on the 4835 studies in the 'PRC' dataset, and information on the subset of the 71 studies in the 'PRC with Selection data' (PRCS) dataset. We used the PRC dataset to assess how representative the PRCS subset was with regard to (1) the observed change in climatic factors over time, and (2) the change in phenotypic traits in response to climatic factors. We define a 'study' as a dataset satisfying our selection criteria for a unique combination of a species, location, climatic factor, phenotypic trait and fitness component. We had more studies than publications because some publications reported data for several species, several climatic factors and/or several phenotypic traits.

**Structure of PRC and PRCS datasets.** The studies in both datasets were predominantly conducted in the Northern Hemisphere (Supplementary Fig. 1). The PRC dataset was heavily biased towards arthropods (88% of studies), with other taxa constituting only a small proportion of the studied species (Supplementary Fig. 2). In contrast, the PRCS dataset was heavily biased towards birds (95%). We found no studies for insects and amphibians that reported annual selection data and satisfied all other inclusion criteria. Among the climatic variables used, temperature dominated both datasets (>70%, Supplementary

Fig. 2); therefore, we focused on the effects of temperature changes in the main text and provide results for precipitation in Supplementary Fig. 3 and Supplementary Note 1. The majority of studies focused on phenological (rather than morphological) traits, with this bias being less pronounced for the PRCS dataset. The median duration of a study was 29 years in the PRCS dataset and 24 years in the PRC dataset (Supplementary Fig. 4).

**Adaptive responses to global warming.** We generally expected that warming temperatures would be associated with an advance in phenological events, because most studies on phenology in our PRC dataset represented early season (spring) events in the Northern Hemisphere, and such events were previously shown to mainly advance with warming temperatures[30,31]. We defined a trait change to be adaptive in response to climate if the climate-driven change in phenotype occurred in the same direction as linear selection. For example, with an increase in temperature over the years, breeding time occurs progressively earlier, with earlier breeding conferring higher fitness (Fig. 1). In contrast, if the trait changed in the direction opposite to selection (e.g. later breeding, despite earlier breeding being favoured), then the response was considered maladaptive[11]. The detected adaptive responses might be due to microevolution, phenotypic plasticity or both. As we used selection differentials that were measured at the phenotypic level, we could not differentiate among these sources.

We conducted separate analyses for PRCS and PRC datasets and, within each of them, for temperature and precipitation. We first quantified the three conditions necessary to infer adaptive responses for each study (Fig. 1). We assessed condition 1 (change in climate over years) with 'model 1'. This linear mixed-effects model predicted the annual values of the climatic variable using the year (modelled as a quantitative variable), thereby estimating the slope of climate on years for each study (Fig. 1b). We assessed condition 2 (change of phenotypic traits with the climatic variable) with 'model 2'. This linear mixed-effects model predicted the mean annual standardized population trait values using the climatic variable (temperature or precipitation, modelled as a quantitative variable), thereby estimating the slope of traits on climate for each study (Fig. 1c). This model also included year as a quantitative variable to account for effects of years on phenotypic traits not mediated by climate. We assessed condition 3 (climate-driven trait changes are associated with fitness benefits) in a two-step procedure. First, we fitted 'model 3'—a linear mixed-effects model that predicted the annual linear selection differentials (weighted by the inverse of their variances) with an intercept, thereby estimating the weighted mean of annual selection differentials (WMSDs) for each study (Fig. 1e, see Methods for details). Second, we plotted the obtained WMSD as a function of the climate-driven trait change over time, calculated as the product of the slopes from conditions 1 and 2, that is, by $\beta_{clim}$ times $\beta_{Trait}$ (Fig. 1f). In this framework, a trait change qualifies as an adaptive response if both WMSD and the trait change over time have the same sign. If their signs differ, then the trait change is maladaptive. We also refitted model 3 using year as a fixed-effect (quantitative) predictor to assess a potential directional change in selection over years (Methods).

Since the measures of phenological responses are sensitive to methodological biases[30], in particular to temporal trends in species abundance[32], we also refitted an extended version of model 2 by additionally including abundance both as a fixed-effect explanatory variable and as an explanatory variable for residual variance (Methods). This model was fitted to a subset of studies for which we could extract abundance data. In all models, we accounted for first-order temporal autocorrelation (if this

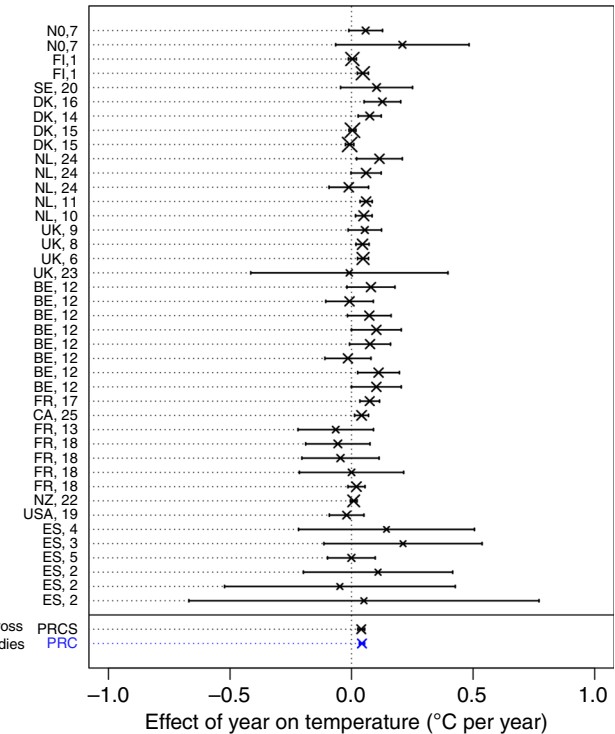

**Fig. 2** Temporal trend in temperature shown for each study in the phenotypic responses to climate with selection (PRCS) dataset. Each study is identified by the publication identity (Supplementary Data 3) and the two-letter country code. Studies are sorted by the decreasing distance of their location from the equator. Bars show 95% confidence intervals and the symbol size is proportional to the study sample size. Dotted lines extending the bars help link the labels to the respective effect sizes. The overall effect sizes calculated across studies in the PRCS dataset (including only studies with selection data, black) and the PRC dataset (including studies with and without selection data, blue) indicate temperature increase over time across studies. Source data are provided as a Source Data file

increased the predictive power of the fitted model), and thus also considered year (modelled as a qualitative variable) as a random effect.

We then performed three meta-analyses to obtain (1) the average slope of climate on years across studies, (2) the average slope of traits on climate across studies and (3) the WMSD across studies. The purpose of these meta-analyses is to provide such average values while accounting for the uncertainty associated with each estimate and for the heterogeneity stemming from variation in study design. All three meta-analyses were performed using mixed-effects models (Methods). We also refitted these models to assess whether the relationships found depended on taxon, type of morphological measure, type of phenological measure, endothermy, fitness component used to measure selection and generation length (Methods). Finally, we compared the proportion of studies showing adaptive responses (i.e. the same sign of WMSD and climate-driven trait change over time) to the proportion of studies showing maladaptive responses (i.e. WMSD and trait change over time differ in their sign) with a binomial test (Methods). We also performed a meta-analysis of the product between WMSD and the sign of the climate-driven trait change over time using a mixed-effects model (Methods).

In line with the recent global temperature increase[33], temperature increased across studies by 0.040 ± 0.007 °C (mean ± SE) per year according to the PRCS dataset (likelihood ratio test [LRT] between the model with and without change in

temperature over years: $\chi^2 = 20.4$, df $= 1$, $p < 0.001$), and by 0.043 ± 0.005 °C per year according to the PRC dataset ($\chi^2 = 41.0$, df $= 1$, $p < 0.001$) (Fig. 2). These rates are slightly higher than those observed in recent meta-analyses that, similarly to our study, are biased toward data from northern latitudes (range 0.03–0.05 °C per year[17,31]). A possible explanation for this discrepancy is that warming rates are higher in recent time series such as ours (Supplementary Fig. 5, median first year in the PRCS dataset $= 1980$, and median study duration $= 29$ years)[31,34].

Consistent with previous work[13,16,17], phenology advanced with increasing temperatures at a rate of −0.260 ± 0.069 standard deviations in the focal trait per degree Celsius (SD per °C) according to the PRCS dataset (LRT between the model with and without change in phenology: $\chi^2 = 11.2$, df $= 1$, $p < 0.001$) and at a rate of −0.248 ± 0.037 SD per °C according to the PRC dataset ($\chi^2 = 22.9$, df $= 1$, $p < 0.001$). In the PRC dataset, the phenological response to temperature varied among taxa (Fig. 3, LRT between the model with and without taxon as a predictor: $\chi^2 = 133.5$, df $= 5$, $p < 0.001$), with the strongest phenological advancement found in amphibians, followed by insects and birds (Supplementary Data 1). This finding is in line with previous research showing that amphibians advanced their phenology faster than other taxa[13,16]. In contrast to Cohen et al.[17], we did not find significant variation in phenological responses among different types of traits (categorized as arrival, breeding and development), either in the PRCS dataset (Supplementary Data 2, LRT between the model with and without the trait type as a predictor: $\chi^2 = 0.5$, df $= 2$, $p = 0.775$) or in the PRC dataset (LRT: $\chi^2 = 0.4$, df $= 2$, $p = 0.809$). Our findings of advancing phenology with warming temperatures were qualitatively unaffected by including abundance, and, although abundance did affect phenological responses, the effects of temperature on phenology were generally larger than those of abundance (Supplementary Fig. 6).

Morphological traits were not associated with temperature in the PRCS (rate of change: 0.060 ± 0.078 SD per °C; LRT: $\chi^2 = 0.6$, df $= 1$, $p = 0.443$) and only marginally associated with temperature in the PRC dataset (rate of change: −0.053 ± 0.029 SD per °C; LRT: $\chi^2 = 3.3$, df $= 1$, $p = 0.068$). Neither endothermy nor type of morphological measure (skeletal vs. body mass) moderated the relationship between morphological traits and temperature (Supplementary Data 2). Our analyses indicated, however, that taxa may moderate the effect of temperature on morphology in the PRC dataset (LRT: $\chi^2 = 4.5$, df $= 1$, $p = 0.11$, Supplementary Data 2), with negative associations on average observed in mammals, and no strong association found in birds (Fig. 3).

Across studies, we found a negative WMSD ($= -0.159 ± 0.061$ SD$^{-1}$) for phenological traits (LRT between the model assuming WMSD is non-zero and the one assuming it equals zero: $\chi^2 = 6.1$, df $= 1$, $p = 0.014$), reflecting higher fitness for earlier-occurring biological events. We also found an indication of the variation in the strength of selection among fitness components (LRT between the model with and without fitness component as a predictor: $\chi^2 = 5.8$, df $= 2$, $p = 0.055$), with the most negative selection acting via recruitment (Fig. 4). We did not find a significant relationship between annual linear selection differentials and years across studies (LRT for phenological traits: $\chi^2 = 0.1$, df $= 1$, $p = 0.764$; LRT for morphological traits: $\chi^2 = 0.5$, df $= 1$, $p = 0.497$, Supplementary Fig. 7). Contrary to selection on phenology, WMSD for morphological traits on average did not differ significantly from zero across studies (WMSD $= 0.044 ± 0.043$ SD$^{-1}$; LRT: $\chi^2 = 1.2$, df $= 1$, $p = 0.268$). We thus did not investigate temporal changes in selection for this trait category.

For phenological traits, negative selection favouring the observed advancing phenology in the context of warming temperature suggests adaptive responses. Accordingly, in 23 out of 38 studies, phenology advanced over time as temperatures

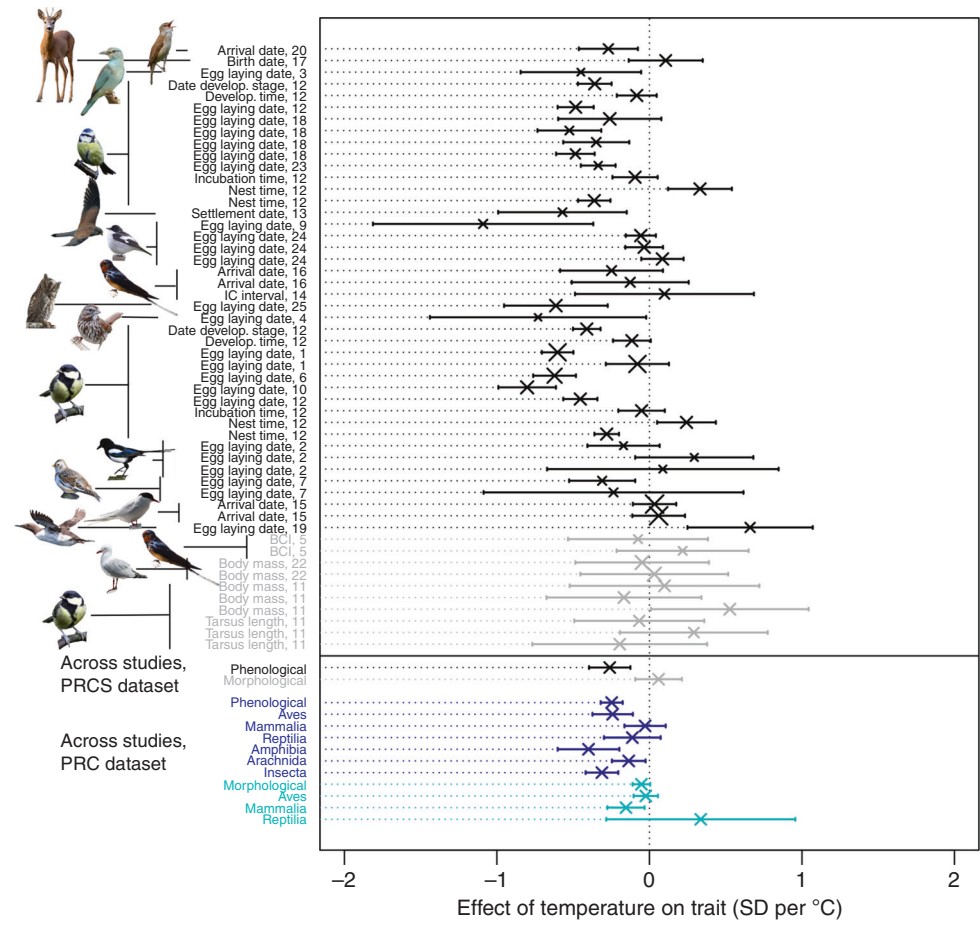

**Fig. 3** Trait changes in response to temperature. For each study in the phenotypic responses to climate with selection (PRCS) dataset, the changes in morphological traits are shown in grey and the changes in phenological traits are shown in black. Each study is identified by the publication identity, the trait and the species. Studies are sorted by trait category (black: phenological; grey: morphological), and within it by species, trait name and publication identity. Overall, phenological traits in both the PRCS dataset (black) and the PRC dataset (dark blue) were negatively affected by temperature. Morphological traits were not associated with temperature in the PRCS (grey) and showed a tendency to a negative association with temperature in the PRC dataset (cyan). In the PRC dataset there was significant variation among taxa in the effect of temperature on phenological (blue) traits, and a tendency to such variation for morphological traits (cyan). See Fig. 2 for legend details. The majority of the species pictures were taken from Pixabay (https://pixabay.com/images/). The exceptions are a picture of red-billed gull (credit: co-author J.A.M.) and four pictures taken from Macaulay library (https://www.macaulaylibrary.org/). Illustration credits for pictures taken from Macaulay library: great reed warbler—Peter Kennerley/Macaulay Library at the Cornell Lab of Ornithology (ML30060261), European pied flycatcher—Suzanne Labbé/Macaulay Library at the Cornell Lab of Ornithology (ML30638911), song sparrow—Steven Mlodinow/Macaulay Library at the Cornell Lab of Ornithology (ML47325951) and Eurasian scops owl—Jon Lowes/Macaulay Library at the Cornell Lab of Ornithology (ML103371221). Source data are provided as a Source Data file

increased, and at the same time negative selection was acting on phenology (studies in quadrant III of Fig. 5a), suggesting adaptive responses. A binomial test revealed a tendency for phenological responses to be more frequently adaptive than maladaptive (mean proportion of studies with adaptive responses = 0.66, $p = 0.07$, Fig. 5). The meta-analysis confirmed the direction of this effect (product of WMSD with the sign of the climate-driven trait change = $0.091 \pm 0.068$), although not reaching significance (Supplementary Fig. 8, LRT: $\chi^2 = 1.9$, df = 1, $p = 0.17$), likely due to high heterogeneity among studies (Higgins $I^2$, i.e. the proportion of total heterogeneity due to between-study variation was 0.999). For morphological traits, which have not changed much over time in response to climate, the proportion of adaptive and maladaptive responses did not differ (Fig. 5, binomial test, mean proportion of studies with adaptive responses = 0.5, $p = 1$).

**Implications for population persistence.** To assess the implications for population persistence of selection acting on

phenology across studies, we used the 'moving optimum' model of Bürger and Lynch[35]. This model, which assumes an optimum phenotype that changes linearly over time due to environmental change, predicts that the lag between the actual population mean phenotype and the optimum should eventually become constant if the population tracks the moving optimum via microevolution (subsequent extensions allowed for phenotypic plasticity, e.g. ref. [36]). This prediction seems valid in our populations since (1) climatic changes are well approximated by a linear trend (Fig. 2) and (2) selection is non-zero and constant over time across studies, as indicated by the lack of a temporal trend in annual linear selection differentials. The Bürger and Lynch[35] model can be used to assess the critical lag behind the optimum, which represents the situation where the population just replaces itself (population growth rate $\lambda = 1$). Comparing the actual to the critical lag provides insight into the expected persistence of populations: if the actual lag is greater than the critical lag, then the population growth rate is lower than 1,

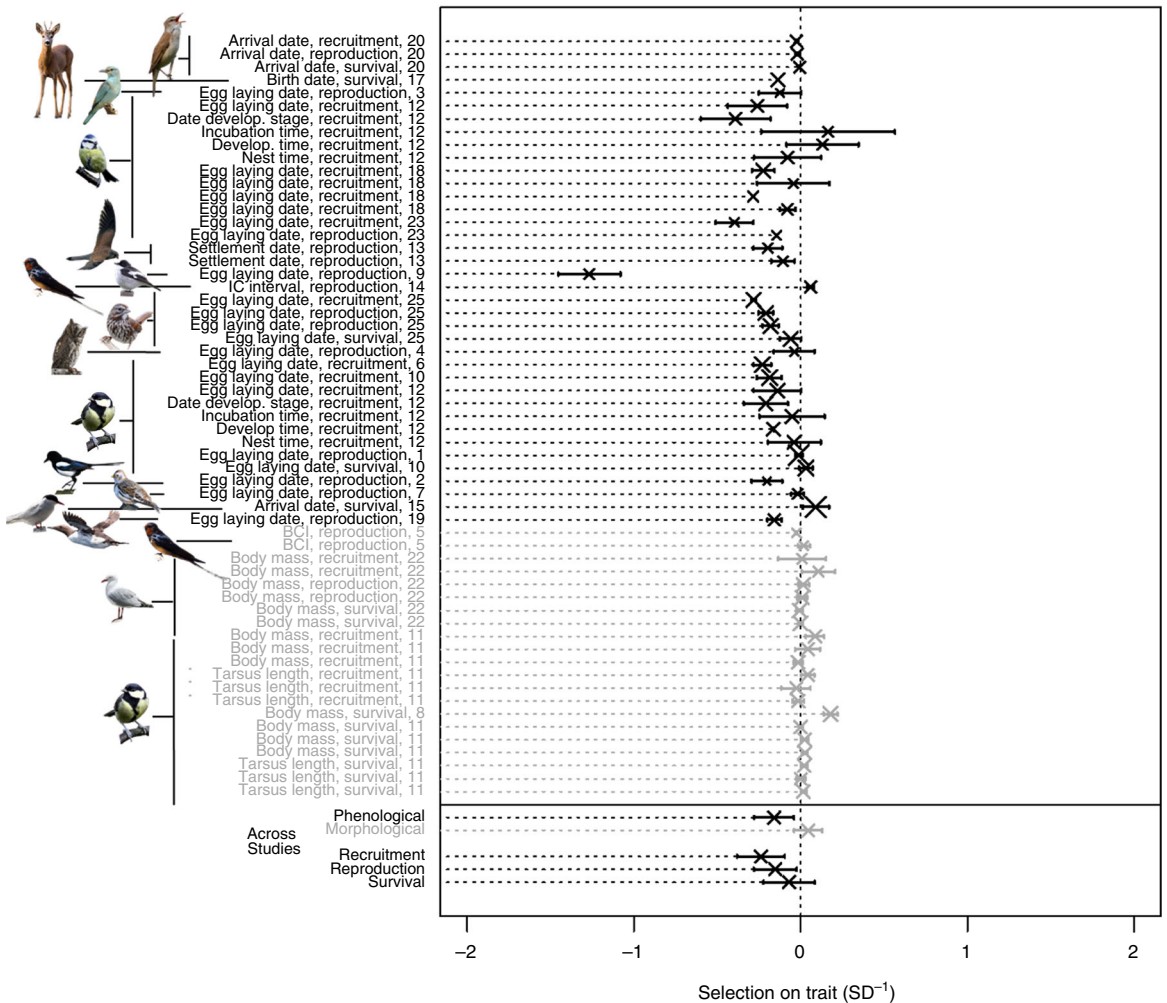

**Fig. 4** Weighted mean of annual selection differentials (WMSDs) for each study. WMSD is shown for phenological (black) and morphological (grey) traits. Each study is identified by the publication identity, the trait, the species and the fitness component. Studies are sorted by trait category (phenological: black; morphological: grey), and within it by species, fitness category and publication identity. Repeated labels correspond to either different locations reported in the same publication, or to measurements on different sexes. Across studies, we found significant negative selection on phenological and no statistically significant selection on morphological traits. There was significant variation in WMSD on phenological traits among fitness components. See Fig. 2 for legend details. Results are robust to the exclusion of the outlier (publication identity 9). The majority of the species pictures were taken from Pixabay (https://pixabay.com/images/). The exceptions are a picture of red-billed gull (credit: co-author J.A.M.) and four pictures taken from Macaulay library (https://www.macaulaylibrary.org/). Illustration credits for pictures taken from Macaulay library: great reed warbler—Peter Kennerley/Macaulay Library at the Cornell Lab of Ornithology (ML30060261), European pied flycatcher—Suzanne Labbé/Macaulay Library at the Cornell Lab of Ornithology (ML30638911), song sparrow—Steven Mlodinow/Macaulay Library at the Cornell Lab of Ornithology (ML47325951) and Eurasian scops owl—Jon Lowes/Macaulay Library at the Cornell Lab of Ornithology (ML103371221). Source data are provided as a Source Data file

meaning substantial extinction risk; otherwise, the populations are assumed to have a negligible extinction risk. The estimation of both the actual and critical lags requires several parameter estimates, which we could not retrieve from the publications behind our data (Methods). However, our numerical analysis of a large parameter space shows that the difference between the actual and critical lags is mostly influenced by two parameters: $\beta$, the strength of directional selection, for which we used the absolute values of our WMSD estimates for each study, and $\omega^2$, the width of the fitness function, for which we did not have study-specific estimates (Fig. 6a–f). We thus applied the Bürger and Lynch[35] model using $\omega^2$ values published for other species[37] together with the study-specific $\beta$ estimates (absolute values of WMSD) and showed that for the populations of 9 out of 13 study species, the actual lag exceeds the critical lag when large values of $\omega^2$ are considered (Fig. 6g). Moreover, the

probability that none of the study species is at risk ($\lambda < 1$) is virtually zero (Supplementary Fig. 9).

## Discussion

To date, the majority of global multi-species studies assessing animal responses to climate change have focused on changes in distribution ranges[3,38,39], whereas phenotypic responses and the extent to which they may be adaptive remain little studied[40]. Moreover, models commonly used to predict species distributions and population viability under climate change usually do not incorporate the potential for species to adapt, often because appropriate data are unavailable to parameterize process-based models[5,41,42]. Our study thus makes an important contribution by focusing on the temporal dimension of species responses to changing environments. We demonstrate that some bird species analysed here seem to respond to warming temperatures by

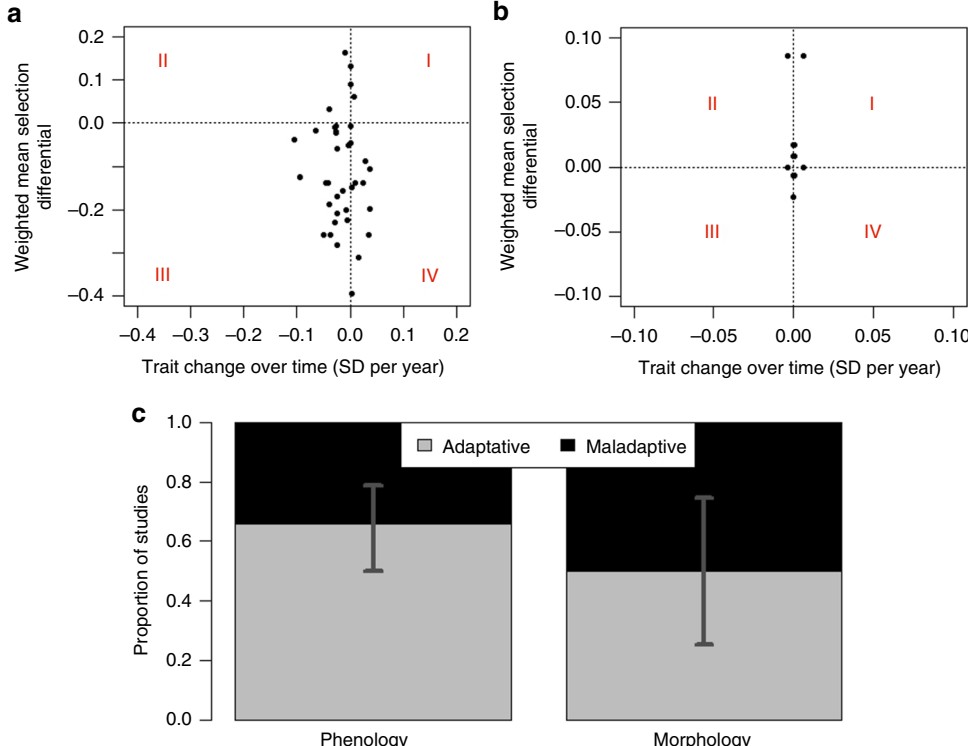

**Fig. 5** Adaptive and maladaptive responses to climate change. **a, b** Weighted mean of annual selection differentials (WMSDs) as a function of the climate-driven phenotypic change over time for **a** phenological and **b** morphological traits. The climate-driven phenotypic change over time is calculated as a product of the slopes from the first two conditions of the framework (the first slope reflects the change in temperature over time and the second slope reflects the change in traits with temperature). Roman numerals shown in red identify four quadrants. Points in quadrant I (upper right) and III (lower left) indicate studies for which phenotypic change over time occurred in the same direction as observed weighted mean annual selection differential, reflecting adaptive responses. Points in quadrants II and IV analogously indicate a maladaptive response. **c** Proportion of studies that showed adaptive and maladaptive phenological and morphological responses. Bars reflect 95% confidence interval (CI). We found a tendency for adaptive phenological responses and no evidence of adaptive responses in morphological traits. Source data are provided as a Source Data file

adaptive advancement of their phenology, emphasizing the possibility of species tracking their thermal niches in situ, which can occur with or without shifts in geographic ranges[43]. However, we did not find evidence for adaptive change in all species, and even populations undergoing adaptive change may do so at a pace that does not guarantee their persistence. We further document variation among fitness components in the strength of selection, with the strongest negative selection stemming from individual variation in recruitment, followed by selection from variation in reproduction and survival. Such strongest selection acting via recruitment and reproduction may point to a mechanism underlying adaptive phenological responses in birds, which is the synchrony of breeding with the availability of resources[7,9,44].

Our findings of adaptive phenological responses to global warming in some bird species, reported here, should not be interpreted over-optimistically. Indeed, perfect adaptation would imply no selection and the significant directional selection observed across studies thus indicates that adaptive responses are imperfect, assuming selection estimates are not consistently biased, for example, see ref. [45]. Furthermore, the lack of a temporal trend in the strength of selection means that, although populations are not perfectly adapted in their phenology, they are not getting more adapted or less maladapted over time as temperatures continue to rise. This result suggests that they are phenotypically tracking a shifting optimum, lagging behind at a constant rate, as predicted by Bürger and Lynch[35]. Our comparisons of the actual vs. critical lags suggest that there is low but non-negligible probability that the degree of maladaptation is large enough for the majority of our study populations to be at

risk. The actual risk of population extinction may in fact be larger because our estimations do not account for several sources of stochasticity[35]. Moreover, our dataset predominantly includes common and abundant species (e.g. *Parus major*, *Cyanistes caeruleus*, *Ficedula hypoleuca*, *Pica pica*) for which collection of selection data is relatively easy. The generality of adaptive phenological responses among rare or endangered species, or those with different life histories, remains to be established[46]. We fear that the forecasts of population persistence for such species will be more pessimistic.

To assess the extent to what animals are able to track climate change, we here used an approach based on selection differentials by testing whether selection over time is significant across studies, and whether it is aligned with the direction of the phenotypic change over time. Alternative approaches exist, for example, the velocity of climate change can be used to assess the expected phenological change that is required to track climate change[18,47]. This approach allowed the authors to demonstrate that, in general, faster phenological shifts occur in regions of faster climate change[18]. Although it would be insightful to compare the results obtained with the approach adopted here and the one based on the velocity of climate, this would only be possible for phenological, and not morphological traits.

Selection and trait change in our analyses were measured at the phenotypic and not the genetic level. Therefore, we cannot determine whether the adaptive phenological responses were due to microevolution or adaptive phenotypic plasticity, nor their relative contributions. Further insights would require differentiating between genetic and environmental components of the

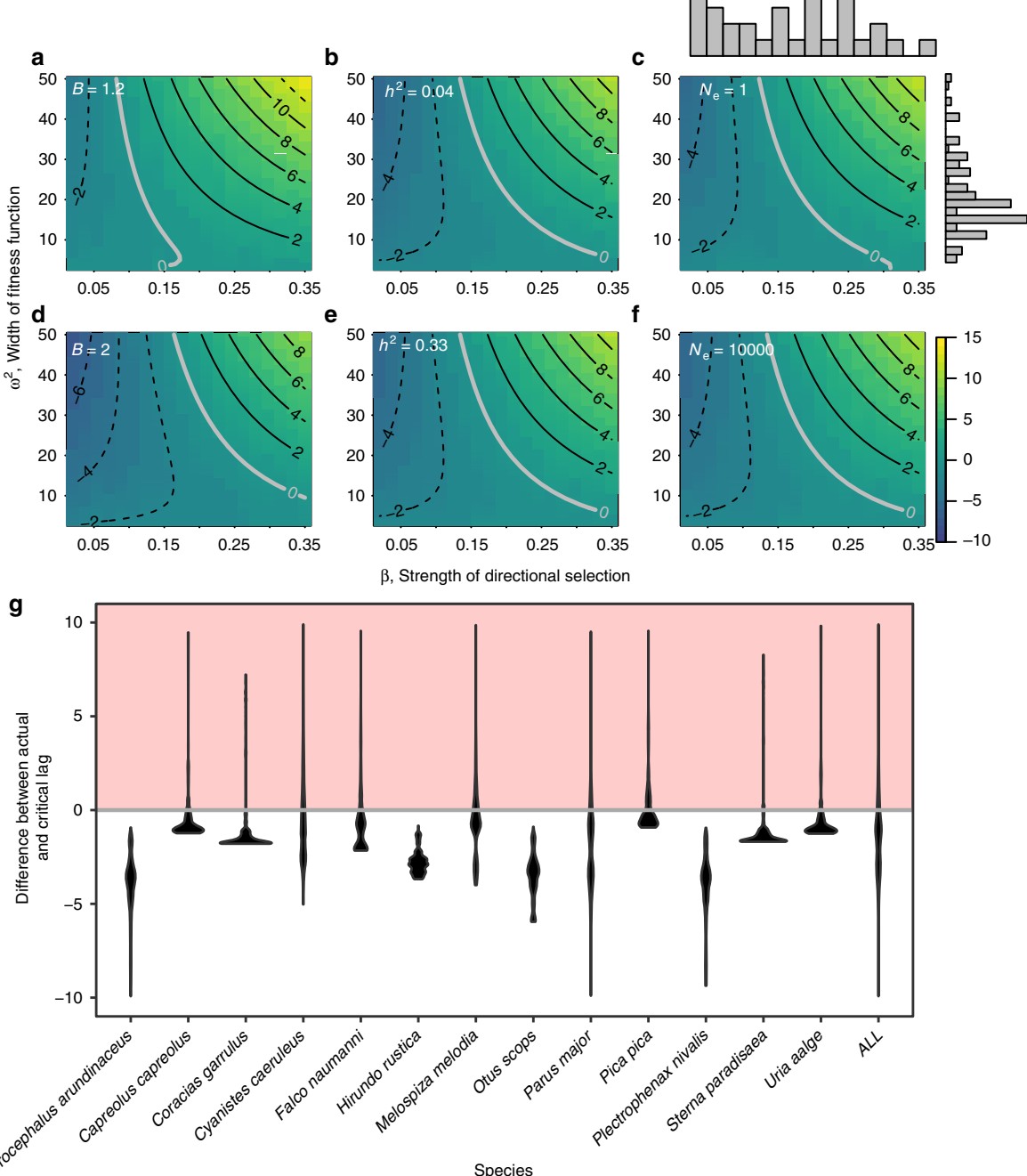

**Fig. 6** Differences between actual and critical lags. **a**–**f** shows differences between actual and critical lags calculated for a range of $\beta$ (linear selection differentials, absolute values) and $\omega^2$ (width of the fitness function) for: **a**, **d** extreme values of parameters $B$ (maximal offspring production), **b**, **e** extreme values of $h^2$ (heritability) and **c**, **f** extreme values of $N_e$ (effective population size), while keeping other parameters at baseline (Supplementary Table 4). **g** Differences between actual and critical lags for species in our dataset (violin plots depict distributions resulting from drawing 1000 $\omega^2$ values and different studies per species). Contour lines show isoclines for the differences (black solid: extinction risk; black dashed: no extinction risk; grey: threshold). Histograms represent distributions of $\beta$ and $\omega^2$ used to produce **g**). Red-shaded area in **g** demonstrates that populations are at risk (i.e. population growth rate < 1)

phenotype and how each relates to the fitness of individuals[10,48]. This could be done by using animal models[11,49], employing common-garden and reciprocal transplant experiments (approaches more suitable for plants, invertebrates and fish) or by combining genetic or genomic and phenotypic information[10,50]. Further, our analyses are correlative and we cannot rule out the possibility that the presumed effects of temperature are in fact due to other or additional environmental variables that correlate with climate, or that selection estimates are biased by

environmental correlations between trait and fitness[51], or do not accurately reflect total selection as a result of being based on incomplete fitness measures.

Similar to the recent global assessments of the climate effects on phenology[17,34] and selection[27], our datasets are heavily biased towards studies from the Northern Hemisphere. Additionally, the majority of the phenological traits in our datasets focus on early season (spring) events. Previous research has shown that early season phenological responses, especially at northern latitudes,

are advancing with warming temperatures[16,31,52], and therefore an advancement of phenological events was our main working assumption. Although the majority of phenological events is reported to advance, delays with warming temperatures have also been recorded[17,53,54]. For example, delay in emergence from hibernation of Columbian ground squirrels was associated with lower fitness, and thus was maladaptive[53]. Similarly, in our study, the majority of maladaptive responses occurred when selection acted in the direction of earlier phenological events, but observed phenological events were delayed over the study period (Fig. 5a). However, whether such delays are generally maladaptive across hemispheres and seasons is unknown. We believe that our proposed framework (Fig. 1) will facilitate answering such questions in the future.

Decrease in body size was suggested to be the third general response of species to global warming, together with changes in phenology and distributions[55,56]. However, evidence for this response is equivocal[14,19,20,55,56]. Our results suggest that inconsistency in findings to date may be explained by different studies focusing on different taxa (e.g. birds[14,20], mammals[57]). Indeed, we found that the association between morphological traits and temperature tends to differ among taxa, with only mammals showing a clear negative association. This finding contrasts with a lack of relationship between temperature and morphology reported for mammals by Meiri et al.[57], potentially because their study periods were longer than ours, and they used a different morphological measure (condyle-basal length). Although our PRC dataset is not exhaustive, our findings of variation among taxa in both phenological and morphological responses highlight the importance of collecting observations on a wide range of taxa.

The assembled PRCS dataset suggests several avenues for further research. For example, currently underappreciated effects of climatic variation on traits and, in turn, on fitness and population viability may be pronounced[58] and our datasets could be used to investigate them. Further, extending this dataset to incorporate vital rates and, ideally, population growth rate would allow for the mapping of environmental changes onto demography via phenotypic traits[44,59,60], and ultimately a better assessment of how trait responses impact population persistence.

Our results are an important first demonstration that, at least in a range of bird species, adaptive phenological responses may partially alleviate negative fitness effects of changing climate. Further work is needed to quantify the extent of such buffering and to broaden the taxonomic scope to determine if this conclusion also applies to species already encountering higher extinction risk for reasons unrelated to climate. The PRC(S) datasets that we assembled should stimulate research on the resilience of animal populations in the face of global change and contribute to a better predictive framework to assist future conservation management actions.

## Methods

**Systematic literature review**. We aimed at assessing adaptive phenotypic responses to climate change across six broad taxa of animals: arachnids, insects, amphibians, reptiles, birds and mammals. We distinguished between two climatic variables: temperature and precipitation. We relied on the authors of the original studies for their expertise and knowledge of the biology of the species and system in the: (1) choice of the appropriate time window over which the annual means of the climatic values were calculated, rather than using a single time window for all species. For instance, if in a bird study the mean temperature over the 2 months preceding nesting was used as an explanatory variable for the timing of egg laying, we used this specific climatic variable; (2) choice of the specific climatic variables, be it air, sea surface or soil temperature, rather than using a single climatic variable across all species; and (3) choice of the spatial scale of the study, so that the measured variables were considered local at that scale. We focused on studies that recorded both changes of at least one climatic variable over time and changes in either morphological or phenological traits for at least one studied species.

Phenological traits reflect shifts in timing of biological events, for example, egg-laying date, antler cast date or mean flight date in insects. Morphological traits reflect the size or mass of the whole body or its parts (e.g., bill length, wing length, body mass).

To assess whether trait changes were adaptive, we only used studies that measured selection on the trait(s) of interest by means of linear selection differentials[29] using one of the following fitness components: recruitment, reproduction, and adult survival. Linear selection differentials for all studies were calculated following Lande and Arnold[29], as the slope of the linear model, with relative fitness (individual fitness divided by mean fitness) as response and the z-transformed trait value as predictor. Only studies that reported SE estimates along with annual linear selection differentials were retained. For the majority of studies, we extracted selection differentials directly from the published studies, and for 12 studies, we calculated them ourselves using the respective individual-level data shared by the authors.

To identify the studies satisfying the above-mentioned criteria, we searched the Web of Knowledge (search conducted on 23 May 2016, Berlin) combining the following keywords for climate change ('climate change' OR 'temperat*' OR 'global change' OR 'precipit'), adaptation ('plastic*' OR 'adapt*' OR 'selection' OR 'reaction norm') and trait category ('body size' OR 'body mass' OR 'body length' OR 'emerg* date' OR 'arriv* date' OR 'breed* date'). For taxa, we used broad taxon names in the first search ('bird*' OR 'mammal*' OR 'arachnid*' OR 'insect*' OR 'reptil*' OR 'amphibia*' OR 'spider*'). Next, to increase the probability of finding the relevant papers, we run the search by using instead of taxa names detailed names below the level of the Class, as follows: ('rodent*' OR 'primat*' OR 'rabbit*' OR 'hare' OR 'mole' OR 'shrew*' OR 'viverrid*' OR 'hyaena' OR 'bear*' OR 'seal*' OR 'mustelid*' OR 'skunk*' OR 'Ailurid*' OR 'walrus*' OR 'pinniped*' OR 'canid*' OR 'mongoos*' OR 'felid*' OR 'pangolin*' OR 'mammal*' OR 'bird*' OR 'flamingo*' OR 'pigeon*' OR 'grouse*' OR 'cuckoo*' OR 'turaco*' OR 'rail*' OR 'wader*' OR 'shorebird*' OR 'penguin*' OR 'stork*' OR 'pelican*' OR 'condor*' OR 'owl*' OR 'hornbill*' OR 'hoopoe*' OR 'kingfisher*' OR 'woodpecker*' OR 'falcon*' OR 'parrot*' OR 'songbird*' OR 'turtle*' OR 'tortoise*' OR 'lizard*' OR 'snake*' OR 'crocodil*' OR 'caiman*' OR 'alligator*' OR 'reptil*' OR 'frog*' OR 'salamander*' OR 'toad*' OR 'amphibia*' OR 'insect*' OR 'beetle*' OR 'butterfl*' OR 'moth*' OR 'mosquito*' OR 'midge*' OR 'dragonfly*' OR 'wasp*' OR 'bee*' OR 'ant*' NOT ('fish*' OR 'water*' OR 'aquatic*')). These detailed taxon names were combined with the same keywords for climate change, adaptation and trait as before. Finally, we joined the unique records from each of these two searches in a single database.

The literature search returned 10,090 publications, 56 of which were retained after skimming the abstracts. Of these 56 publications, 23 contained the data necessary to assess the three conditions required to infer adaptive responses and were used for assembling the final dataset (PRCS dataset). In cases where several publications reported on the same study system (same species in the same location, measuring the same traits and selection via exactly same fitness components), we retained the publication that reported data for the longest time period. We assembled the PRCS dataset by directly extracting the data from the identified 23 publications wherever possible, or by contacting the authors to ask for the original data. Data were extracted either from tables directly or from plots by digitizing them with the help of WebPlotDigitiser or the metagear package in R[61]. In the process of contacting the authors, one research group offered to share relevant unpublished data on two more species, adding two more studies to the dataset, totalling 25 publications. The PRCS dataset consisted of 71 studies containing data on annual values of climatic factors, annual phenotypic trait values and annual linear selection differentials for 17 species in 13 countries (Supplementary Data 3).

The remaining 33 publications from the originally selected 56 (58 with the shared unpublished data considered as two publications) did not report data on selection, but presented data on the annual values of climatic factors and mean population phenotypic traits, totalling 4764 studies that covered 1401 species. We retained these studies and combined them together with the 71 studies in the PRCS dataset to assemble the 'PRC' dataset (Supplementary Data 3). With the PRC dataset, we did not aim for comprehensive coverage of the literature published on the topic. Instead, we used this larger PRC dataset to verify whether the smaller PRCS dataset was representative in terms of climate change over time and trait change in response to a climatic factor. A flowchart showing the numbers of studies included at each stage of the systematic literature review is given in Supplementary Fig. 10.

**Assessing whether the responses are adaptive**. Separate analyses were conducted for the PRCS and PRC datasets and, within each of them, for temperature and precipitation. All analyses were conducted using linear models as no deviation from linearity was detected by visual inspection of the relations between (1) year and climate, (2) trait and climate and (3) selection and year for each study (Supplementary Figs. 11–15). First, we assessed for each study condition 1 necessary to infer adaptive responses (i.e. the extent to which the climatic variable changed directionally over years). To this end, we fitted for each study a mixed-effects model with the climatic variable as the response and the year as a fixed covariate, taking into account temporal autocorrelation (as random effect):

$$\text{Clim}_t = \alpha + \beta_{\text{Clim}} \times \text{Year}_t + \varepsilon_t + \varepsilon, \tag{1}$$

where Clim is the quantitative climate variable, Year the quantitative year covariate,

$t$ the time, $\varepsilon_t$ is a Gaussian random variable with mean zero and following an AR1 model over years, and $\varepsilon$ is an independent Gaussian random variable with mean zero and variance representing the residual variance of the study. $\beta_{Clim}$ is the regression coefficient reflecting the slope of the climatic variable on the year for the study (Fig. 1b). To avoid overfitting, we refitted the same model without the AR1 structure and retained, for each study, the model structure leading to the lowest marginal AIC[62]. This approach was applied to all the models fitted to each study (i.e. to assess conditions 2 and 3, and change of selection across years, as described below).

We assessed condition 2, the relation between the trait and the climatic variable, separately for phenological and morphological traits. For this, we fitted a mixed-effects model for each study with mean annual population trait values as a response and the climatic variable and the year as fixed effects. Year was included as a quantitative predictor in this model to account for the effects of variables other than the considered climatic variable, which had changed with time and could have affected the trait. Examples of such variables are any environmental alterations, such as land use change and succession but also other climatic variables, which potentially could have affected the trait, but for which we did not have data. In this model, we took into account temporal autocorrelation in the response variable and weighted the residual variance by the variance of the response variable (i.e. the reported squared SE of the mean annual population trait values) to account for between-year variation in uncertainty associated with mean annual population trait values. Prior to fitting the models we z-transformed trait values (i.e. subtracted the mean and divided by their reported standard deviation) to later compare the effect of the climatic variable on different traits. Accordingly, we also transformed the weights of the residual variance by dividing the reported SEs by the SD of the mean annual population trait values per study. The fitted model was:

$$\text{Trait}_t = \alpha + \beta_{Trait} \times \text{Clim}_t + \gamma \times \text{Year}_t + \varepsilon_t + \varepsilon, \qquad (2)$$

where Trait is the mean phenotypic trait (z-scaled across the years within the study), Clim the quantitative climate covariate, Year the quantitative year covariate, $t$ the time, $\varepsilon_t$ is a Gaussian random variable with mean zero and following a AR1 model over years, and $\varepsilon$ is an independent Gaussian random variable with mean zero and variance proportional to the estimated variance of the mean phenotypic trait (which depends on $t$). $\beta_{Trait}$ is the regression coefficient reflecting the slope of the trait on the climatic variable for the study (Fig. 1c).

We assessed condition 3 of whether the trait change was associated with fitness benefits in a two-step procedure. In the first step, we fitted for each study an intercept-only mixed-effects model with annual linear selection differentials as a response. We allowed for temporal autocorrelation and weighted the residual variance by the variance of the annual linear selection differentials (i.e. the reported squared SE of the annual selection differentials). The fitted model was:

$$\text{Sel}_t = \alpha + \varepsilon_t + \varepsilon, \qquad (3)$$

where Sel is the estimate of the yearly linear selection differential, $t$ is the time, $\varepsilon_t$ is a Gaussian random variable with mean zero and following an AR1 model over years, and $\varepsilon$ is an independent Gaussian random variable with mean zero and variance proportional to the estimated variance of the annual linear selection differential (which depends on $t$). The intercept $\alpha$ describes a non-zero mean of the autoregressive process. The predictions from the fitted model (Sel$_t$), including the random effect, are estimates of annual linear selection differentials, and their inverse-variance weighted average is termed 'weighted mean annual selection differential', WMSD (Fig. 1e). The variance used in weighting is the prediction variance. The SE of the WMSD is deduced from these weights and from the covariance matrix of the predictions (see source code of the function extract_effects () in our R package 'adRes' for details).

In the second step, to assess whether the response is adaptive, we considered WMSD in combination with the slopes obtained for the previous two conditions, as follows: we defined a trait change to be adaptive in response to climate if the climate-driven change in phenotype occurred in the same direction as linear selection. In contrast, if the climate-driven change in phenotype occurred in the direction opposite to selection, then the response was considered maladaptive. We measured the climate-driven change in phenotype as the product of the slopes obtained for conditions 1 and 2. A WMSD estimate of zero indicates a lack of selection[11]. A WMSD of zero together with no trait change could indicate a stationary optimum phenotype, and a WMSD of zero together with a significant change in trait could indicate that a moving optimum phenotype is perfectly tracked by phenotypic plasticity (a negligible WMSD could also imply a flat fitness surface, i.e. no fitness penalty for deviating from the optimum). To assess whether the trait is adaptive, we plotted for each study the WMSD against the product of slopes extracted from conditions 1 and 2 (Fig. 5). The studies qualify as adaptive if their WMSD has the same sign as the product of slopes assessing conditions 1 and 2.

We also fitted a modified version of the model specified in Eq. (3) to assess a potential temporal (linear) change in the annual linear selection differentials over years. To this end, for each study we fitted a mixed-effects model that accounted for temporal autocorrelation. We weighted the residual variance by the variance of the annual linear selection differentials (i.e. the reported squared SE) to account for uncertainty in the estimates of annual selection differentials. The fitted model was:

$$\text{Sel}_t = \alpha + \beta_{Sel} \times \text{Year}_t + \varepsilon_t + \varepsilon, \qquad (4)$$

where Sel is the estimate of the yearly linear selection differential, Year the quantitative year covariate, $t$ the time, $\varepsilon_t$ a Gaussian random variable with mean zero and following an AR1 model over years, and $\varepsilon$ is an independent Gaussian random variable with mean zero and variance proportional to the estimated variance of the yearly linear selection differential (which depends on t). $\beta_{Sel}$ is the regression coefficient that corresponds to the slope of the annual linear selection differentials on the year for the study.

**Meta-analyses.** To demonstrate general responses across species and locations, we require each of the three conditions necessary to infer adaptive responses to be met consistently across studies, for example, that, on average, temperature increased over time, warmer temperatures were associated with advancing phenology and advancing phenology corresponded to fitness benefits (i.e. negative selection on phenological traits given the two above-mentioned conditions are satisfied). To test for such general trends in adaptive responses across studies, we fitted three mixed-effects meta-analyses to the PRCS dataset, two for the first two conditions and the third to assess whether WMSD differed from zero. We tested the third condition in two ways. First, we performed a binomial test to compare the proportion of studies exhibiting adaptive (i.e. same sign for WMSD and the climate-driven trait change over time) vs. maladaptive (i.e. WMSD and the climate-driven trait change over time differ in their signs) responses to climate change. Second, we performed a mixed-effects meta-analysis similar to the three other ones.

First, we assessed whether, across studies, the values of the climatic factor changed with time by using the slope of a climatic factor on year (obtained from the mixed-effects models of condition 1 for each study, see above) as response (i.e. effect size in meta-analysis terminology), and study identity and publication identity as qualitative variables defining random effects influencing the intercept. Second, to assess whether climate change was associated with trait changes across studies, we used the slope of the z-transformed trait on the climatic factor (obtained from the mixed-effects models of condition 2 while accounting for the effect of year on the trait) as response and study identity and publication identity as qualitative variables defining random effects influencing the intercept. We fitted separate models for phenological and morphological traits, because our dataset contained fewer studies of the latter compared to the former. Since morphological traits included either measures of body mass or size (e.g. wing, tarsus and skull length), we tested whether the effect of temperature depended on the type of measure by including it as a fixed-effect covariate with three levels (body mass, size and body condition index; we distinguished body condition index from the two other levels as it has elements of both of them). Analogously, we assessed whether the effect of temperature on phenology depended on the type of phenological measure used, by including it as a fixed-effect covariate with three levels, similarly to Cohen et al.[17]: arrival, breeding/rearing (e.g. nesting, egg laying, birth, hatching) and development (e.g. time in a certain developmental stage, antler casting date). Third, to assess whether, across studies, traits were under positive or negative selection during the study period, we used as response the WMSD values obtained from the mixed-effect models for the first step assessing condition 3. In this model, we also used study identity and publication identity as qualitative variables defining random effects influencing the intercept. We tested whether selection depended on generation length and differed among fitness components by including these latter variables as fixed effects in the model. Generation length was extracted from the literature, mainly using the electronic database of BirdLife International. Similarly, to assess whether across studies there was a directional linear change in the annual linear selection differentials over time, we fitted a mixed-effects model using as response the slopes of the annual linear selection differentials on time (obtained with Eq. (4)). This model included study identity and publication identity as qualitative variables defining random effects influencing the intercept. Finally, to assess whether responses were on average adaptive, we also ran a mixed-effects meta-analytic model using as response the product of WMSD with the sign of the climate-driven trait change over time. We included study identity and publication identity as qualitative variables defining the random effects in this model. We fitted separate models for phenological and morphological traits to test whether both WMSD and the product of WMSD with the sign of the climate-driven trait change differed from zero.

For each type of climatic variable (temperature and precipitation) in the PRC dataset, we fitted two mixed-effects meta-analyses, analogous to the mixed-effects meta-analytic models we ran on the PRCS dataset. With these meta-analyses we assessed whether across the studies (1) there was a directional change in the climatic values over time and (2) traits were affected by the climatic variable. As responses (i.e. effect sizes) in these models, we used the slopes extracted for each study from the respective mixed-effects models fitted analogously to those used for the PRCS dataset (see section above). For both morphological and phenological traits, we assessed whether the effect of climate on traits differed among taxa by including taxon as a fixed effect. For morphological traits, we also assessed whether the responses to climate differed among endothermic and ectothermic animals, by including endothermy as a fixed effect in the model.

All data analyses were conducted in R version 3.5.0[63] and implemented in the R package 'adRes', which is provided for the sake of transparency and reproducibility. Mixed-effects models for each study and mixed-effects meta-analytic models were fitted using restricted maximum likelihood (ML) with the spaMM package version 2.4.94[64]. For each meta-analytic mixed-effects model, we conducted model

diagnostics by inspecting whether the standardized residuals deviated from a normal distribution and whether there were any patterns in standardized residuals when regressed on the predictor. Model diagnostics were satisfactory for all models. We assessed the significance of fixed effects and intercepts with asymptotic likelihood ratio $\chi^2$ tests by comparing the model with a given effect vs. the model without the effect, both fitted using ML in spaMM.

We assessed the amount of heterogeneity among studies in our meta-analyses using commonly recommended approaches[65]. In particular, we tested whether the total amount of heterogeneity ($Q$) was statistically significant and estimated Higgins $I^2$ and $H^2$ (ref. [65]). Higgins $I^2$ reflects the proportion of total heterogeneity due to between-study variation (i.e. random effects) and ranges from 0 to 1. A value of 0 means that heterogeneity is due to within-study variation exclusively, whereas a value of 1 indicates that heterogeneity is due to between-study variation. This metric is, therefore, comparable among different meta-analyses. $H^2$ is a ratio showing the proportion of observed heterogeneity in relation to what would be expected under the null hypothesis of homogeneity. For example, a value of 2 means that there is twice as much variation as would be expected if no between-study variation were present (i.e. $H^2 = 1$). We found that the amount of heterogeneity differed among the two datasets and tested models, with moderate heterogeneity for models testing conditions 1 and 2 and considerable heterogeneity for models testing condition 3 (Supplementary Note 1, Supplementary Table 1).

We also tested for the evidence of publication bias by (1) visual inspection of the funnel plots and (2) Egger's test[65]. No evidence of small-study effect (that may be an indication of publication bias) was found for effect sizes used to test all three conditions in the PRCS dataset (Supplementary Figs. 16–17, Supplementary Note 1).

**Sensitivity analyses**. One study in the meta-analytical models testing conditions 2 and 3 for phenology in the PRCS dataset appeared to be an outlier (ref. [66], Figs. 3 and 4). Therefore, we also re-ran the models without this study (Supplementary Tables 2 and 3). Additionally, the PRCS dataset contained mainly studies on bird species, and only one for a mammal species[67]. To test how sensitive the results were to inclusion of this taxon, we re-ran the analyses after excluding the study for mammal species. The main findings were qualitatively unaffected by excluding either the only mammal study or an apparent outlier from the PRCS dataset, or both (Supplementary Tables 2 and 3).

Measures of phenological responses are known to be sensitive to methodological biases[30], in particular to temporal trends in abundance of the sampled species[32]. Indeed, if species abundance increases over time, the probability of recording earlier events increases (especially if phenology is measured as the first occurrence date), meaning that species abundance can affect mean of the distribution of a phenological trait. For the same reason, the variance around the mean phenological response may be sensitive to population size, with higher variance at lower population sizes. To assess the sensitivity of our results to potential changes in population sizes over time, we fitted a heteroskedastic model in which population abundance was included both as a fixed-effect explanatory variable and as an explanatory variable for the model of the residual variance. This model is an extension of the models specified in Eq. (2) that were used to assess condition 2. This model was fit to each study for which abundance data were available and where the duration of the study was at least 11 years (28 out of 42 studies). The fitted model was:

$$\text{Trait}_t = \alpha + \beta_{\text{Trait}} \times \text{Clim}_t + \gamma \times \text{Year}_t + \delta \times \text{Abund}_t + \varepsilon_t + \varepsilon, \qquad (5)$$

where Trait is the mean phenotypic trait (z-scaled across the years within each study), Clim the quantitative climate covariate, Year the quantitative year covariate, Abund the quantitative covariate for species abundance, $t$ the time, $\varepsilon_t$ is a Gaussian random variable with mean zero and following an AR1 model over years, and $\varepsilon$ is an independent Gaussian random variable with mean zero and variance proportional to the estimated variance of the mean phenotypic trait (which depends on $t$). $\beta_{\text{Trait}}$, $\gamma$ and $\delta$ are regression coefficients that correspond to the slopes of the trait on the climatic variable, the trait on year and the trait on the abundance for the study, respectively. To avoid overfitting, we refitted the same model without the AR1 structure and retained, for each study, the model structure leading to the lowest marginal AIC, analogously to how it was done for the models fitted to assess conditions 1, 2 and 3.

The results were qualitatively unaffected by the inclusion of abundance in the models: the rate of advancement in phenology estimated across the studies was $-0.346 \pm 0.102$ SD per °C (LRT between the model with and without change in phenology: $\chi^2 = 8.4$, df $= 1$, $p = 0.004$) when including abundance and $-0.335 \pm 0.088$ SD per °C without abundance (LRT: $\chi^2 = 9.19$, df $= 1$, $p = 0.002$; for comparison's sake both models were fitted to a subset of studies for which abundance data were available). We also found that although abundance affects phenology, its effects were on average smaller compared to the effects of temperature (Supplementary Fig. 6).

**Implications for population persistence**. To assess implications of our findings for population persistence, we first calculated the actual lag (Lag) between the observed mean phenotype and the optimum by following Estes and Arnold (2007)[37]:

$$\text{Lag} = \beta \cdot \left( \omega^2 + \sigma_p^2 \right), \qquad (6)$$

where $\beta$ is a standardized linear selection differential, $\omega^2$ a width of the fitness function, and $\sigma_p^2$ phenotypic variance (here scaled to 1). This lag is expected to be asymptotically constant when the optimum is shifting at a constant rate and under other assumptions detailed in Bürger and Lynch[35].

Next, we calculated the critical lag (Lag$_{\text{crit}}$) between the phenotype and the optimum following Bürger and Lynch[35] (see also ref. [68]):

$$\text{Lag}_{\text{crit}} = k_c / s, \qquad (7)$$

where $k_c$ is the critical rate of environmental change and $s$ is a measure for the strength of selection, $s = \sigma_g^2 / (\sigma_g^2 + V_s)$, with $\sigma_g^2$ genetic variance and $V_s$ the strength of stabilizing selection around the optimum, calculated as $V_s = \omega^2 + \sigma_e^2$, with $\omega^2$ the width of the fitness function and $\sigma_e^2$ environmental variance in the trait.

The critical lag reflects a situation where the population can just replace itself (population growth $\lambda = 1$) and comparison between the actual and the critical lag provides insight into the persistence of populations. If the actual lag is larger than the critical lag, then the population growth rate is lower than 1, implying substantial extinction risk. We therefore focused our numeric analyses on this difference between the actual lag and the critical lag. Note that this analysis relies on strong assumptions described in the original papers (e.g. heritability is considered in the estimation of the critical lag but not in that of the actual lag, and that the reverse is true for plasticity), but more accurate methods, for example, ref. [36], would require detailed data that are not available for most of our study populations.

In the first step, we assessed the difference between the actual and the critical lag across a range of parameter values (Supplementary Table 4) to study the sensitivity of the difference to known (i.e. estimated) or unknown parameters. This analysis revealed that the results are most sensitive to the values of $\omega^2$ and $\beta$ (Fig. 6a–f). Therefore, in the second step, we computed the difference among the observed and critical lags for each study in our PRCS dataset (Fig. 6g), by (1) using as $\beta$ the absolute values of the estimates of WMSD for each study and (2) drawing 1000 random $\omega^2$ per study. We drew these random values from the empirical distribution of $\omega^2$ estimates provided in Fig. 7 from Estes and Arnold[37].

**Reporting summary**. Further information on research design is available in the Nature Research Reporting Summary linked to this article.

## Data availability

The PRC dataset, containing metadata on each study as well as coefficients estimated for each step of the framework (Fig. 1) per each study, is available as Supplementary Data 4. We also make available the raw data for a subset of studies for which we received the consent of data owners (4819 out of 4835 studies). These data are available as part of the R package 'adRes' (www.github.com/radchukv/adRes), which implements the complete workflow of this study and provides functions to be used to conduct similar analyses on new data in the future. The source data underlying Figs. 1b, c, 1e, f, 2, 3, 4 and 5a–c are provided as a Source Data file.

## Code availability

The complete workflow implementing each step of the proposed framework (Fig. 1) is available as the R package 'adRes' (www.github.com/radchukv/adRes). The functions included in the package allow conducting the analyses as described in this study and can be applied to either the shared dataset that is included in the R package or to a new dataset formatted in the same way.

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

## Acknowledgements

We are grateful to many researchers who collected field data and kindly shared them. Without such contributions this study would have been impossible. In particular, we would like to thank the 'Wicken Fen ringing group', Mats Björklund, Richard du Feu, Philip W. Hedrick, Loeske Kruuk, Andrew G. McAdam, Josephine Pemberton, Lisa Schwanz, William Sydeman, Dakota McCoy, Rory Telemeco, Chris Thorne, Marco Rughetti, Wolf Blanckenhorn and Tim Sparks. This study was initiated by the workshop series of the Leibniz Forschungsverbund (LVB) Biodiversität 'Species adaptations to global change—a comprehensive risk analysis' 2014 and 2015. This work was supported by funds from: European Research Council (ERC-2013-StG-337365-SHE to A.Ch., ERC-2013 -AdG-339092-E-Response to M.E.V., ERC-2014-StG-639192-ALH to T.R.), Ministry of Economy and Competitivity, Swedish Research Council (621-2014-5222 to B.H.), Spanish Research Council (CGL-2016-79568-C3-3-P to J.C.S.), US National Science Foundation (DEB-1242510 to F.J., DEB-0089473 to F.S.D.), the Academy of Finland (project 265859 to T.E.) and the US Department of Energy (Award Number DE-FC09-07SR22506 to the University of Georgia Research Foundation to D.S.).

## Author contributions

V.R., A.Co. and S.K.-S. designed the study; V.R. conducted literature search and gathered the data; V.R. and A.Co. analysed the data with significant inputs from F.R.; V.R., A.Co. and T.R. conducted numerical analysis based on a theoretical model; T.R., C.T., M.v.d.P., A.Ch., A.W., C.H., P.Ad., F.A., M.P.A., P.Ar., J.M.A., J.B., A.B., S.B., S.J.G.H., J.C., N.D., F.d.L., A.A.D., N.J.D., H.D., T.E., J.J., J.M., J.F., F.F., A.E.G., B.H., M.H., D.H., H.K., I.F., J.G.M., J.-B.M., J.A.M., K.S.B., M.M.-M., M.-O.R., A.P.M., A.M., A.O., D.P., T.K., M.P., D.S., J.C.S., C.S., B.G.S., T.H., K.T., M.T., C.E.T., J.T., P.P., P.T., B.S., E.M., F.J., F.S.D., M.E.V. and S.R.B. contributed data or input at the LVB workshops; V.R. wrote the manuscript; all authors edited the manuscript and made valuable scientific contributions throughout the writing process.

## Additional information

**Competing interests:** The authors declare no competing interests.

Viktoriia Radchuk[1], Thomas Reed[2], Céline Teplitsky[3], Martijn van de Pol[4], Anne Charmantier[3], Christopher Hassall[5], Peter Adamík[6], Frank Adriaensen[7], Markus P. Ahola[8], Peter Arcese[9], Jesús Miguel Avilés[10], Javier Balbontin[11], Karl S. Berg[12], Antoni Borras[13], Sarah Burthe[14], Jean Clobert[15], Nina Dehnhard[16], Florentino de Lope[17], André A. Dhondt[18], Niels J. Dingemanse[19], Hideyuki Doi[20], Tapio Eeva[21], Joerns Fickel[1,22], Iolanda Filella[23,24], Frode Fossøy[25,26], Anne E. Goodenough[27], Stephen J.G. Hall[28], Bengt Hansson[29], Michael Harris[14], Dennis Hasselquist[29], Thomas Hickler[30], Jasmin Joshi[31,32], Heather Kharouba[33], Juan Gabriel Martínez[34], Jean-Baptiste Mihoub[35], James A. Mills[36,37], Mercedes Molina-Morales[17], Arne Moksnes[24], Arpat Ozgul[38], Deseada Parejo[17], Philippe Pilard[39], Maud Poisbleau[16], Francois Rousset[40], Mark-Oliver Rödel[41], David Scott[42], Juan Carlos Senar[13], Constanti Stefanescu[23,43], Bård G. Stokke[24,25], Tamotsu Kusano[44], Maja Tarka[29], Corey E. Tarwater[45], Kirsten Thonicke[46], Jack Thorley[47,48], Andreas Wilting[1], Piotr Tryjanowski[49], Juha Merilä[50], Ben C. Sheldon[51], Anders Pape Møller[52], Erik Matthysen[7], Fredric Janzen[53], F. Stephen Dobson[54], Marcel E. Visser[4], Steven R. Beissinger[55], Alexandre Courtiol[1,57] & Stephanie Kramer-Schadt[1,56,57]

[1]Leibniz Institute for Zoo and Wildlife Research (IZW), Alfred-Kowalke-Straße 17, 10315 Berlin, Germany. [2]School of Biological, Earth and Environmental Sciences, University College Cork, Cork T23 N73K, Ireland. [3]CEFE UMR 5175, CNRS – Université de Montpellier – Université Paul-Valéry Montpellier – EPHE, 1919 route de Mende, 34293 Montpellier Cedex 5, France. [4]Department of Animal Ecology, Netherlands Institute of Ecology (NIOO-KNAW), P.O. Box 506700 AB Wageningen, The Netherlands. [5]School of Biology, Faculty of Biological Sciences, University of Leeds, Leeds LS2 9JT, UK. [6]Department of Zoology, Palacký University, tř. 17. listopadu 50, 771 46 Olomouc, Czech Republic. [7]Evolutionary Ecology Group, University of Antwerp, Universiteitsplein 1, 2610 Wilrijk, Belgium. [8]Swedish Museum of Natural History, P.O. Box 5000710405 Stockholm, Sweden. [9]Department of Forest and Conservation Sciences, 2424 Main Mall, Vancouver V6T 1Z4 BC, Canada. [10]Department of Functional and Evolutionary Ecology, Experimental Station of Arid Zones (EEZA-CSIC), Ctra de Sacramento s/n, 04120 Almería, Spain. [11]Department of Zoology, Faculty of Biology, University of Seville, Avenue Reina Mercedes, 41012 Seville, Spain. [12]Department of Biological Sciences, University of Texas Rio Grande Valley, Brownsville 78520 TX, USA. [13]Museu de Ciències Naturals de Barcelona, P° Picasso s/n, Parc Ciutadella, 08003 Barcelona, Spain. [14]Centre for Ecology and Hydrology, Bush Estate, Penicuik EH26 0QB, UK. [15]Station of Experimental and Theoretical Ecology (SETE), UMR 5321,

CNRS and University Paul Sabatier, 2 route du CNRS, 09200 Moulis, France. [16]Behavioural Ecology and Ecophysiology Group, University of Antwerp, Universiteitsplein 1, 2610 Wilrijk (Antwerp), Belgium. [17]Department of Anatomy, Cellular Biology and Zoology, University of Extremadura, 06006 Badajoz, Spain. [18]Lab of Ornithology, Cornell University, Ithaca, NY 14850, USA. [19]Behavioural Ecology, Department of Biology, Ludwig-Maximilians University of Munich, Großhaderner Str. 2, Planegg-Martinsried 82152, Germany. [20]Graduate School of Simulation Studies, University of Hyogo, 7-1-28 Minatojima-minamimachi, Kobe 650-0047, Japan. [21]Department of Biology, University of Turku, Turku FI-20014, Finland. [22]Institute for Biochemistry and Biology, Potsdam University, Karl-Liebknecht-Strasse 24-25, 14476 Potsdam, Germany. [23]CREAF, 08193 Cerdanyola del Vallès, Spain. [24]CSIC, Global Ecology Unit CREAF-CSIC-UAB, Bellaterra 08193, Spain. [25]Norwegian Institute for Nature Research (NINA), P.O. Box 5685 Torgarden, 7485 Trondheim, Norway. [26]Department of Biology, Norwegian University of Science and Technology (NTNU), Høgskoleringen 5, 7491 Trondheim, Norway. [27]School of Natural and Social Sciences, University of Gloucestershire, Swindon Road, Cheltenham GL50 4AZ, UK. [28]Estonian University of Life Sciences, Kreutzwaldi 5, 51014 Tartu, Estonia. [29]Department of Biology, Lund University, 22362 Lund, Sweden. [30]Senckenberg Biodiversity and Climate Research Center (BiK-F), Senckenberganlage 25, 60325 Frankfurt/Main, Germany. [31]Biodiversity research/Systematic Botany, University of Potsdam, Maulbeerallee 1, Berlin 14469, Germany. [32]Institute for Landscape and Open Space, HSR Hochschule für Technik, Oberseestrasse 10, Rapperswil 8640, Switzerland. [33]Department of Biology, University of Ottawa, Ontario K1N 6N5, Canada. [34]Departamento de Zoologia, Facultad de Ciencias, Universidad de Granada, 18071 Granada, Spain. [35]Sorbonne Université, Muséum National d'Histoire Naturelle, CNRS, CESCO, UMR 7204, 61 rue Buffon, 75005 Paris, France. [36]10527A Skyline Drive, Corning, NY 14830, USA. [37]3 Miromiro Drive, Kaikoura 7300, New Zealand. [38]Department of Evolutionary Biology and Environmental Studies, University of Zurich, Zurich 8057, Switzerland. [39]LPO Mission Rapaces, 26 avenue Alain Guigue, 13104 Mas-Thibert, France. [40]ISEM, Université de Montpellier, CNRS, IRD, EPHE, Montpellier 34095, France. [41]Leibniz Institute for Evolution and Biodiversity Science, Museum für Naturkunde, Invalidenstrasse 43, 10115 Berlin, Germany. [42]Savannah River Ecology Laboratory, University of Georgia, Aiken, SC 29802, USA. [43]Natural History Museum of Granollers, Francesc Macià, 52, 08401 Granollers, Spain. [44]Department of Biological Sciences, Tokyo Metropolitan University, 1-1 Minami-Osawa, Hachioji-shi, Tokyo 192-0397, Japan. [45]Department of Zoology and Physiology, University of Wyoming, 1000 E University Avenue, Laramie, WY 82071, USA. [46]Research Domain 1 'Earth System Analysis', Potsdam Institute for Climate Impact Research (PIK), P.O. Box 60 12 03, Telegrafenberg A31, Potsdam D-14412, Germany. [47]Imperial College London, Silwood Park Campus, Buckurst Road, Ascot SL5 7PY, UK. [48]Department of Zoology, University of Cambridge, Downing Street, Cambridge CB2 3EJ, UK. [49]Institute of Zoology, Poznan University of Life Sciences, Wojska Polskiego 71C, 60-625 Poznań, Poland. [50]Organismal and Evolutionary Biology Research Programme, Ecological Genetics Research Unit, Faculty Biological and Environmental Sciences, University of Helsinki, 00014 Helsinki, Finland. [51]Edward Grey Institute, Department of Zoology, University of Oxford, Oxford OX1 3PS, UK. [52]Ecologie Systématique Evolution, Université Paris-Sud, CNRS, AgroParisTech, Université Paris-Saclay, 91405 Orsay Cedex, France. [53]Department of Ecology, Evolution, and Organismal Biology, Iowa State University, Ames, IA 50011, USA. [54]Department of Biological Sciences, Auburn University, Auburn, AL 36849, USA. [55]Department of Environmental Science, Policy and Management and Museum of Vertebrate Zoology, University of California, Berkeley 94720 CA, USA. [56]Department of Ecology, Technische Universität Berlin, 12165 Berlin, Germany. [57]These authors jointly supervised this work: Alexandre Courtiol, Stephanie Kramer-Schadt.

