## [Peer Review File · Nature Communications]

Reviewers' Comments:

Reviewer #1:

Remarks to the Author:

The manuscript represents an impressive and rigorous attempt to assess adaptive responses to climate change. The methodology goes beyond studies inferring phenotypic shifts are adaptive by testing for adaptive responses using three clearly defined criteria. The dataset is necessarily taxonomically, regionally, and otherwise limited, but provides a robust basis for the study. The manuscript makes a substantial contribution to understanding adaptation in response to climate change, but some aspects require clarification and figures could be improved. I particularly appreciated the analyses of temporal change in selection and lags.

Broad comments

The discussion of whether phenotypic shifts arise from genetic change or plasticity (L294) is adequate to address the issue, but the issue should be mentioned earlier in the manuscript. An earlier statement should clarify that phenotypic shifts are treated as microevolution in the selection and lag analysis.

I found it difficult to follow how many cases met all three conditions for phenotypic adaptive responses. It would be useful to report this information in the text (or at least do so more prominently). I took the language in L452 to suggest that the analyses focus on whether the conditions are met on average? It would also be informative to modify the figures to report this information. For example, labels or symbols in figure 4 could indicate whether all three conditions were met.

The manuscript would benefit from some figure improvements:

For figure 1, it would be nice to see an actual example.

I appreciated the plots corresponding to conditions 1 and 2 in the supplement, but it would be helpful to indicate whether slopes are significant in each panel. I would also find it informative to be able to examine the three conditions together for each study. I would thus suggest that the panels be reordered such that each column is a condition and each row is a study. Why isn't data for condition 3 shown? It seems that seeing the data would be particularly valuable to understand differences in the magnitude and direction of selection across years.

How are the studies ordered in figure 3 and 4? I would again suggest considering combining the three conditions as columns in a single figure (but understand that it would require some repetition and the figure could be hard to read). Regardless of whether the conditions are combined, shading the background of rows would help match the data to the labels. It would be helpful to specify in figure 4 that repeated labels indicate multiple populations.

The argument that constant selection across years suggests that species are lagging behind an optimum at a constant rate is fairly convincing, but I wondered whether comparing observed to expected phenological shift directly would bolster the claim (as has been done in the velocity of climate change analyses of range shifts). The season shift metric mapped by Burrows et al. (2011, *Science*) and used in an analysis by Poloczanska et al. (2013, *Nature Climate Change*) wouldn't match the study durations unless recalculated, but could provide a rough indication of how well species are tracking phenology. Perhaps not worth the analysis effort.

L313 I agree with the authors that there are promising avenues for further research using their dataset. I thus urge them to make their dataset publically available (at least after a time lag). The

data availability section implies that the PRC dataset will only be released as coefficients. Raw data seems essential to replicating the analyses and should be released.

Specific comments

L171 Was there a minimum duration defined for inclusion in the database?

L359 Might the choice of phenology terms emerg* OR arriv* OR breed* have biased the dataset toward birds? Where other terms considered such as those related to development? But I expect the bird bias would persist regardless.

L545 I'm unclear in figure 5 why a histogram of Beta values is shown and the legend indicates the distribution is used to produce B. Aren't actual Beta values for each study used?

L454 Why directionality in the conditions? Any cases where delayed phenology might be adaptive?

Fig S6 It would also be informative to report SD in selection across years.

Reviewer #2:

Remarks to the Author:

The goal of this manuscript is to evaluate whether there is sufficient evidence to support the hypothesis that climate change is causing adaptive change in the traits of species. I believe that the authors correctly point out three criteria necessary to support this hypothesis and that we lack studies that simultaneously assess all three criteria. Given the fact that our climate is indeed changing and that it is unclear whether climate change has produced adaptive changes within species, I believe that this work has great merit and would appeal to many readers. To complete this work the authors conducted a vigorous literature review to compile data and performed statistical analyses that evaluate these criteria. I believe that the work completed is compelling and I have no substantial concerns in regards to the analyses completed. This study reveals that phenological traits but not morphological traits are responding to climate change in an adaptive way. These results are very interesting and have important policy implications and grist for future studies.

My biggest concern with this manuscript involves the way in which the methods (particularly statistical analyses) are written within the main part of the text. I found them misleading and unclear. I initially found myself wondering why the authors performed the analyses the way they did but after reading the detailed methods after the manuscript my concerns were allayed. In particular, I do not believe that the description of the models in the main text (primarily between lines 179-184) reflect an accurate summary of the detailed methods. Critical aspects of the models included in the detailed methods are missing in the shortened aspect to such a degree that I first questioned the validity of the results reported. I believe that the authors will need to provide a more accurate and persuasive summary of the statistical models in the main text so as to not distract readers into thinking there are problems with the study because some may not bother to get into the detailed methods at that point). I outline some of the things that entered my head when I first read lines 179-184 below but after I read the detailed methods I see my understanding of what the authors actually did was not correct and the concerns I outline below are unwarranted. Furthermore, the text describing the process of compiling the data in the main text was rather confusing without also reading the detailed methods. I offer a suggestion on how to revise this so that it is clearer on its own while the detailed methods can remain as is to give the more complete picture.

Line 148: Without reading the detailed methods which appear later, the text describing the number of studies and the rationale for the two data sets is unclear at this point. This paragraph could be made

clearer if the authors just made clear that their literature search resulted in 1) 5348 studies (representing 1401 species in 23 countries) that contained information on phenotypic responses to climate change and 2) 93 studies (representing 17 species in 13 countries) that contained all of the information required to assess whether responses were adaptive (then issue a reference to the detailed methods to outline how this information was obtained). Then name these two data sets and include the description already provided by the authors about the proposed usage of each data set.

Lines 179-184: For model 1 why was the independent effect of study not included in the model? Doesn't the absence of an independent effect of "study" presume that the y-intercept of the model (i.e., the estimated value of the climate variable at time 0) is the same in all studies? This presumes that climate did not vary in space at some point in time. Also, why presume that there is not a general trend in climate which would require the inclusion of the "year" effect? It would not surprise me if the way that climate changes through time varies among studies but I am perplexed why the authors are ignoring these other effects? Similarly, why does model 2 ignore the independent effects of climate and study on population trait values? I was glad to see model 3 basically included the independent effects of study and year but why was the interaction ignored? Couldn't the directional change in selection over years also differ among studies? For example, couldn't directional changes ramp up more rapidly in polar areas where climate change has become most pronounced while directional changes occur more slowly in tropical areas where climate change is less pronounced?

Reviewer #3:

Remarks to the Author:

This paper is interesting and an important contribution to the literature on adaptive responses to climate change. The scope of the meta analysis is comprehensive, and the insights it generate are interesting. The paper is accessible, well written, and an enjoyable read. However, like all meta analyses, there are limitations that need to be addressed either methodologically or at least acknowledged in the methods and interpretation of the results.

Major concerns:

The structure of the meta-analytic and mixed effects models seems appropriate, and inclusion of autocorrelated error terms is warranted. My major concerns with the analyses involve (category 1) elements beyond the control of the authors, and (category 2) elements the authors can control, or at least try to control:

Category 1) There's likely implicit bias against publication of non-significant results, possibly for a couple of reasons. Authors may be disinclined to attempt publication of non-significant results, or such results may have a hard time finding their way into the literature, especially in journals tracked by Web of Knowledge (the source used in this paper). There's nothing the authors can do about this, but they should at least acknowledge this as a possible source of bias in their literature search.

Category 2) Variation or trends in phenotypic and phenological traits may be due to variation or trends in environmental factors such as temperature (the focus here), variation or trends in abundance/population density (not considered here), or some interaction of these. One of the co-authors on this paper has been an advocate for assessing artifactual effects of changes in abundance on detection of phenological trends. For instance, significant trends toward earlier timing may in some instances be an artifact of population trends if the distribution of the phenological trait is approximately normal and the variance about the mean is sensitive to population size. In addition to such effects, trends in abundance or density might drive or contribute to trend in phenotypic and phenological traits through density dependent competition. This isn't acknowledged in the methods

description, but should be.

Ideally, the authors should tackle this directly, perhaps with a subset of the meta data if abundance time series accompany only some of the trait and phenology meat data. This would necessitate including estimates of abundance or density as predictors. The most statistically robust approach would involve allowing for interaction between climate and abundance/density. For instance, climate trends can drive trends in abundance, with knock-on effects on traits and timing of events. The autocorrelated error term may be partitioned to account for this, for instance using a state space model, because any significant autocorrelation might be ascribed to a density dependent process. Alternatively, a structural equation modeling approach might be necessary, with latent effects for abundance/density.

Minor comments:

- 1) State in abstract the actual number of studies used in the meta-analysis so as to avoid misrepresenting the scope of the analyses. Most readers will only read the abstract.
- 2) Line 171: Should report median, not average.
- 3) Lines 193-195: In some cases, phenological delay might be adaptive or neutral. Alternatively, Cohen et al. (ref. 17) in their recent NCC paper suggest detection of phenological delays might also in some instances be related to sampling error (specifically, short annual records). At any rate, these considerations should also be given some space in the text.
- 4) Lines 203-210, especially lines 209-201: The authors should test this directly, rather than speculate. Compare the rate of warming to period of study and between urban and non-urban areas, or something along those lines. For instance, try regressing estimated rate of warming against the first, median, or last year of the study.
- 5) Lines 216-220. Taxonomic variation is not surprising, but the authors should also include phenological trait in their mixed effects model. Presumably, even within taxa, there is some variation in phenological traits represented in the meta data. This should be accounted for as in the recent meta analysis by Post et al. 2018 Sci. Rep..
- 6) Lines 362-363: What is the taxonomic distribution across these 23 papers? Is that what Table S2 shows? If not, this information should be detailed somewhere in the main text or supplemental material.
- 7) Lines 406-408: Year was included as a predictor variable to account for influences of factors other than climate. I understand the intent, but without identifying these, how can you reject "random" variation, or even effects of changes in sampling effort or methodology through time?
- 8) Lines 439-441: A trait change is defined as adaptive in response to directional climate change if the phenotype change occurred in the same direction as selection. This strikes me as circular. It's difficult to be convinced by this since there's no way to determine whether any such association with climatic variables is direct or masking some underlying association with, for instance, simultaneous trends in abundance/density, as explained above under "major concerns".
- 9) Lines 454-455: The emphasis/operating assumption throughout this paper is that phenological

advances in response to warming are adaptive. There's also abundant evidence for phenological delays in some traits and species (e.g., later season events, such as end of growing season or onset of leaf coloration in deciduous forests - Andrew Richardson's group has published numerous papers on this). In some cases, it's presumed this may relate to moisture limitation with increasing temperature, or to indirect effects of cloudiness and solar irradiance in tropical systems (e.g., Pau et al. NCC 2013). Such trends might also be adaptive, no? Alternatively, advancing phenology may in some instances be maladaptive or even neutral. Certainly, this must be possible even for different traits within species.

Reviewers' comments:

Reviewer #1:

The manuscript represents an impressive and rigorous attempt to assess adaptive responses to climate change. The methodology goes beyond studies inferring phenotypic shifts are adaptive by testing for adaptive responses using three clearly defined criteria. The dataset is necessarily taxonomically, regionally, and otherwise limited, but provides a robust basis for the study. The manuscript makes a substantial contribution to understanding adaptation in response to climate change, but some aspects require clarification and figures could be improved. I particularly appreciated the analyses of temporal change in selection and lags.

Our answer: *We thank the reviewer for his/her positive feedback.*

Broad comments

The discussion of whether phenotypic shifts arise from genetic change or plasticity (L294) is adequate to address the issue, but the issue should be mentioned earlier in the manuscript. An earlier statement should clarify that phenotypic shifts are treated as microevolution in the selection and lag analysis.

Our answer: *We agree with the reviewer and we have now specified earlier in the manuscript (LL 187-189) that our analyses do not differentiate between microevolution and adaptive phenotypic plasticity when assessing whether the responses are adaptive.*

I found it difficult to follow how many cases met all three conditions for phenotypic adaptive responses. It would be useful to report this information in the text (or at least do so more prominently). I took the language in L452 to suggest that the analyses focus on whether the conditions are met on average? It would also be informative to modify the figures to report this information. For example, labels or symbols in figure 4 could indicate whether all three conditions were met.

Our answer: *While we agree that it is technically possible to focus on individual studies separately, we decided not to follow this proposition (although we now provide p-values for each study in supplementary figures, as the reviewer asked for; see below). The reason is that we fear that such an approach would be misleading as it would go against the general philosophy of meta-analysis. Indeed, as the reviewer points out, the goal of meta-analyses is precisely to assess whether on average – i.e. across the studies – a given effect is significant (in our case whether each of three conditions is met across the studies). The goal is not to focus on individual studies, because such an approach may be under-powered and even biased. As a thought experiment, because meta-analyses combine the statistical power of multiple studies, it is for example possible that a meta-analysis reveals a significant pattern while all individual (under-powered) studies may not.*

Counting the number of significant studies, as proposed by the reviewer, is thus inappropriate as it conflates questions of sample size with effect size. Instead, we thus chose to rely on effect sizes (and their SE) from each study to draw inference about the general response across studies. Our approach is traditional in this respect and follows the recommendations of proponents of meta-analysis (e.g. Borenstein et al. 2009; Koricheva et al. 2013). We made sure to articulate this point better in the manuscript (LL 220-224).

The manuscript would benefit from some figure improvements:

For figure 1, it would be nice to see an actual example.

Our answer: As suggested, we now have used one of the studies for the demonstration of our conceptual framework in Fig. 1. More precisely, we have used the study 49, from Wilson et al. 2007 – the publication identity 25 in our Supplementary Table S3.

I appreciated the plots corresponding to conditions 1 and 2 in the supplement, but it would be helpful to indicate whether slopes are significant in each panel. I would also find it informative to be able to examine the three conditions together for each study. I would thus suggest that the panels be reordered such that each column is a condition and each row is a study. Why isn't data for condition 3 shown? It seems that seeing the data would be particularly valuable to understand differences in the magnitude and direction of selection across years.

Our answer: We have now added the significance level to the plots in Figs. S10-S12 and we have also added the supplementary Figs. S13-S14 with the raw data on selection (condition 3) which illustrates the magnitude and direction of selection across years.

Regarding the request to reorder the panels in such a way that each condition is a column and each row is a study, we decided against it because it would put the emphasis on individual studies, which we would rather avoid for the reasons stated above. Further, such a plot would be very difficult to read because to test condition 1 we must take a single slope of temperature over years per each study location. Yet, we often have several slopes to be plotted per single study location for conditions 2 and 3 if several traits were measured at that same location (e.g. phenological or morphological traits that were tested separately; and / or several fitness components that were reported by one publication). Such a plot thus would have slopes for condition 1 shown repeatedly for the slopes corresponding to conditions 2 and 3.

How are the studies ordered in figure 3 and 4? I would again suggest considering combining the three conditions as columns in a single figure (but understand that it would require some repetition and the figure could be hard to read). Regardless of whether the conditions are combined, shading the background of rows would help match the data to the labels. It would be helpful to specify in figure 4 that repeated labels indicate multiple populations.

Our answer: The studies in Fig. 3 are ordered by the trait category (first phenological, then morphological), and within each trait category by the species name, the trait name and the publication identity. In Fig. 4, similarly, the studies are ordered by the trait category, and within each trait category by the species name, the fitness category and the publication identity. We have now specified this in the figure legends. As explained in the above comment, we would rather not plot the results for all three conditions in one figure, as that would make such a figure hard to read and prone to misinterpretation. To improve the ability to match the effect sizes with the respective labels we have added dotted lines linking each effect size to its label in Figs. 2-4 (and, analogously, updated Supplementary Figs. S3 & S8). Also, we now explain in the legend to Fig. 4 that repeated labels correspond to either different locations investigated in the same publication, or to measurements on different sexes.

The argument that constant selection across years suggests that species are lagging behind an optimum at a constant rate is fairly convincing, but I wondered whether comparing observed to expected phenological shift directly would bolster the claim (as has been done in the velocity of climate change analyses of range shifts). The season shift metric mapped by Burrows et al. (2011, Science) and used in an analysis by Poloczanska et al. (2013, Nature Climate Change) wouldn't match the study durations unless recalculated, but could provide a rough indication of how well species are tracking phenology. Perhaps not worth the analysis effort.

Our answer: We are thankful to the reviewer for making us aware of these relevant studies, we now cite Poloczanska et al. (2013) in the revised manuscript. In fact, the temporal change in climate is already quantified in our framework by the analysis of condition 1. We think that regressing the observed phenotypic lags vs the seasonal climate shifts, similarly to how it was done by Poloczanska et al. (2013) would not further help to answer the questions we are addressing in this study, because it would only show whether the shifts in climate correlate with the shifts in phenotype (which we already infer from testing conditions 1 and 2). Here, we instead aim to understand how the phenotypic shifts due to climate change relate to fitness. Specifically, we investigate the implications of the existing phenotypic shifts ('lags') to the persistence of populations, by comparing the observed phenotypic lag with the one that would be critical for population persistence (using the 'moving optimum' model of Burger & Lynch (1995)). We tried to improve the description of this model in the revised manuscript version (LL 603-630).

L313 I agree with the authors that there are promising avenues for further research using their dataset. I thus urge them to make their dataset publically available (at least after a time lag). The data availability section implies that the PRC dataset will only be released as coefficients. Raw data seems essential to replicating the analyses and should be released.

Our answer: We now release the code implementing the complete analyses as an R package (www.github.com/radchukv/adRes). Together with this package we also provide the raw data for those studies whose data holders agreed to this (4819 out of 4835 studies). Overall, we still provide metadata and the effect sizes extracted per study for each of the three conditions (as Supplementary

Table S8), thus enabling the replication of the results reported in this manuscript and facilitating subsequent meta-analyses.

Specific comments

L171 Was there a minimum duration defined for inclusion in the database?

Our answer: Yes, only studies with a minimum duration of 6 years were included in the database. This is mentioned on LL 143.

L359 Might the choice of phenology terms emerg* OR arriv* OR breed* have biased the dataset toward birds? Where other terms considered such as those related to development? But I expect the bird bias would persist regardless.

Our answer: Thank you for raising this important point. While the use of emerg and arriv* could have biased our search towards bird studies, the use of breed* is common in other animal studies (those on mammals, reptiles and amphibians). In fact, at the beginning of the study we have varied our search keywords widely, investigating how the use of the terms would affect the number of hits. For example, we also had 'phenol' and 'morphol' included as keywords, however their inclusion did not increase the number of hits much compared to the currently used keywords, and therefore we retained the searching strategy described in the paper.*

L545 I'm unclear in figure 5 why a histogram of Beta values is shown and the legend indicates the distribution is used to produce B. Aren't actual Beta values for each study used?

Our answer: We thank the reviewer for spotting this mistake in the legend of figure 5. As explained in the Methods (LL 627-629), in order to produce the results summarized in Fig. 5B we used Beta estimates for each study and only the distribution of omega values was sampled (1000 random values per study). We have now corrected the figure legend.

L454 Why directionality in the conditions? Any cases where delayed phenology might be adaptive?

Our answer: We hypothesized that global warming is predominantly associated with phenological advances (and not delays) because we focus predominantly on early (spring) events and the majority of the studies in our database is coming from the Northern Hemisphere. Early season (spring) events, which are the focus of this study, especially in the Northern Hemisphere, were previously reported to mainly advance with climate change (Brown et al. 2016; Post et al. 2018). This is why we also expect to find this result in our data. We have now added a paragraph in the Discussion (LL 337-342) to recognize this caveat.

Although advances in phenology prevail in the literature, phenological delays have also been reported (Lane et al. 2012; Miles et al. 2017; Cohen et al. 2018). However, phenological delays were

mainly reported for the late season events (primarily autumn). We are unaware of the studies looking at whether phenological delays are adaptive, but theoretically this is possible. For example, the phenological delay in a specialist herbivore may be adaptive if it occurs in response to the delay in phenology of its food source. In the Discussion we now highlight that phenological delays have also been reported and that future research should address the question of whether delays in the phenology are adaptive or not (LL 342-347).

Fig S6 It would also be informative to report SD in selection across years.

Our answer: We are not absolutely clear what the reviewer means by 'SD in selection across years'. If what is meant is between-year variation in selection, then this is what we show by bars on Fig. 4. Indeed, the plotted bars on Fig. 4 reflect the 95% CI for WMS, calculated across years by fitting the mixed-effects model as detailed in Methods (eq. 4). On the other hand, if the reviewer means CI of the selection differentials measured each year, we now have added such information in Figs. S13-S14 (as explained above).

Reviewer #2:

The goal of this manuscript is to evaluate whether there is sufficient evidence to support the hypothesis that climate change is causing adaptive change in the traits of species. I believe that the authors correctly point out three criteria necessary to support this hypothesis and that we lack studies that simultaneously assess all three criteria. Given the fact that our climate is indeed changing and that it is unclear whether climate change has produced adaptive changes within species, I believe that this work has great merit and would appeal to many readers. To complete this work the authors conducted a vigorous literature review to compile data and performed statistical analyses that evaluate these criteria. I believe that the work completed is compelling and I have no substantial concerns in regards to the analyses completed. This study reveals that phenological traits but not morphological traits are responding to climate change in an adaptive way. These results are very interesting and have important policy implications and grist for future studies.

Our answer: We thank the reviewer for his/her positive feedback and encouraging support.

My biggest concern with this manuscript involves the way in which the methods (particularly statistical analyses) are written within the main part of the text. I found them misleading and unclear. I initially found myself wondering why the authors performed the analyses the way they did but after reading the detailed methods after the manuscript my concerns were allayed. In particular, I do not believe that the description of the models in the main text (primarily between lines 179-184) reflect an accurate summary of the detailed methods. Critical aspects of the models included in the detailed methods are missing in the shortened aspect to such a degree that I first questioned the validity of the results reported. I believe that the authors will need to provide a more accurate and persuasive summary of the statistical models in the main text so as to not distract readers into

thinking there are problems with the study because some may not bother to get into the detailed methods at that point).

Our answer: We are glad that the reviewer does not challenge the analysis itself nor its full-fetched description. We have now tried to summarize more clearly our methods in the main text (LL 190-220); see below for more details.

I outline some of the things that entered my head when I first read lines 179-184 below but after I read the detailed methods I see my understanding of what the authors actually did was not correct and the concerns I outline below are unwarranted. Furthermore, the text describing the process of compiling the data in the main text was rather confusing without also reading the detailed methods. I offer a suggestion on how to revise this so that it is clearer on its own while the detailed methods can remain as is to give the more complete picture.

Our answer: We thank the reviewer for pointing this out and for suggestions to improve the description; we followed them as detailed below.

Line 148: Without reading the detailed methods which appear later, the text describing the number of studies and the rationale for the two data sets is unclear at this point. This paragraph could be made clearer if the authors just made clear that their literature search resulted in 1) 5348 studies (representing 1401 species in 23 countries) that contained information on phenotypic responses to climate change and 2) 93 studies (representing 17 species in 13 countries) that contained all of the information required to assess whether responses were adaptive (then issue a reference to the detailed methods to outline how this information was obtained). Then name these two data sets and include the description already provided by the authors about the proposed usage of each data set.

Our answer: We revised this part of our manuscript as proposed by the reviewer (LL 150-157). Note that we realized that the reported number of studies included in each dataset was incorrect, this is now corrected.

Lines 179-184: For model 1 why was the independent effect of study not included in the model? Doesn't the absence of an independent effect of "study" presume that the y-intercept of the model (i.e., the estimated value of the climate variable at time 0) is the same in all studies? This presumes that climate did not vary in space at some point in time. Also, why presume that there is not a general trend in climate which would require the inclusion of the "year" effect? It would not surprise me if the way that climate changes through time varies among studies but I am perplexed why the authors are ignoring these other effects? Similarly, why does model 2 ignore the independent effects of climate and study on population trait values? I was glad to see model 3 basically included the independent effects of study and year but why was the interaction ignored? Couldn't the directional change in selection over years also differ among studies? For example, couldn't directional changes

ramp up more rapidly in polar areas where climate change has become most pronounced while directional changes occur more slowly in tropical areas where climate change is less pronounced?

Our answer: Several concerns raised by the reviewer may stem from the lack of clarity of the brief summary of our statistical method, which we have now tried to address. Let us here reply to each concern individually:

- *Lack of the study effect in model 1: this is a misunderstanding, we did consider a different intercept value for each study as shown in equation 1 (in the old manuscript version), expressed by the index i relating to the study i . We fit a linear (mixed-effects) regression to obtain both an intercept and a slope of the effect of years on climate for each study (hence the index " i " in the equation for these terms). Then, we compute the mean of all slopes obtained using a meta-analytical framework. Since we are only interested in the effect of year on climate, there is no need to also compute the average intercept across all studies. Indeed, because all slopes are initially estimated jointly with their corresponding intercept, our method does not presume that "climate did not vary in space at some point in time".*
- *Lack of year effect in model 1: this is not a mistake; including a simple term Year in our first model in addition to the interaction term would lead to a strictly identical fit (same likelihood, same number of degrees of freedom) in comparison to our interaction-only model, but would lead to a different parameterization which would be less straightforward to handle in the follow-up meta-analysis. Indeed, for the two alternative parameterizations (our original one and the one proposed by the reviewer), the equivalent of one intercept and one slope is computed for each study. Following the parameterization proposed by the reviewer (and under the default settings in R i.e. "treatment contrast") the coefficient for the term Year would represent the slope corresponding to the effect of years in one specific study (the reference level for the factor 'study') and the model fit would also provide $N-1$ (with N being the number of studies) other slopes that would be expressed as the difference between the effect of years for a given study and the effect of years for the reference study. In contrast, omitting the effect of Year per se and considering only the intercept plus the interaction between year and study implies that the N slopes obtained are readily representing the effect of years for each study. So we did consider that "the way that climate changes through time varies among studies". This is precisely the result that we depict in our Figure 2.*
- *Lack of independent effects of climate and study on population trait values in model 2: this misunderstanding combines the two misunderstandings just mentioned above, which stems from unclear description of the methods in the main manuscript (old version). The effect of climate is considered and assumed to differ between studies (see index " i " in $\beta_i \times Clim_{i,t}$ from equation 2 in the old version of the manuscript) and the effect of the study on the trait is considered via one intercept value per study (α_i in equation 2 in the old version of the manuscript). See previous answers for an explanation.*
- *Lack of interaction between study and year in model 3: the model 3 indeed estimates study-specific intercepts, which reflect the weighted mean selection over time per each study. We did not include the interaction between the study and year, because we found no temporal change in selection across the studies (see Section 'Assessing temporal change in selection' in*

Methods, LL 585-601 and the respective results reported on LL 265-266 and Supplementary Fig. S8).

All these comments led us to rethink some of the modelling decisions we had made. In particular, we have now realized that estimating the same level of auto-correlation between years across all studies is an assumption that makes little biological sense. The same is true for the estimation of the residual variance which we had constrained across studies in some of the models. We have thus chosen to relax these assumptions and have revised the methods accordingly. One important consequence is that we now fit 3 models (to assess each condition) for each study and no longer have to cope with the complexity of estimating between-study effects within single models for all studies (as implemented before). Therefore, our methodology is now much easier to explain (LL 190-212 in revised MS). Another difference with the former analysis is that we no longer consider in meta-analytical models the species and the study location as random effects. Indeed, those showed high collinearity with the study and the publication ID, respectively. Therefore, the partitioning of variance was highly unstable and we prefer the entire variance to be characterised by a single term for simplicity. We have updated all figures and tables based on the new model fits. Importantly, all our results remain qualitatively similar, which demonstrates the robustness of the results to slight modifications in modelling assumptions.

Reviewer #3:

This paper is interesting and an important contribution to the literature on adaptive responses to climate change. The scope of the meta analysis is comprehensive, and the insights it generate are interesting. The paper is accessible, well written, and an enjoyable read. However, like all meta analyses, there are limitations that need to be addressed either methodologically or at least acknowledged in the methods and interpretation of the results.

Our answer: We thank the reviewer for his/her feedback and constructive suggestions. We addressed all comments below.

Major concerns:

The structure of the meta-analytic and mixed effects models seems appropriate, and inclusion of autocorrelated error terms is warranted.

Our answer: We are glad that both reviewer 2 and 3 agree with our statistical methodology.

My major concerns with the analyses involve (category 1) elements beyond the control of the authors, and (category 2) elements the authors can control, or at least try to control:

Category 1) There's likely implicit bias against publication of non-significant results, possibly for a couple of reasons. Authors may be disinclined to attempt publication of non-significant results, or such results may have a hard time finding their way into the literature, especially in journals tracked by Web of Knowledge (the source used in this paper). There's nothing the authors can do about this, but they should at least acknowledge this as a possible source of bias in their literature search.

Our answer: We are thankful to the reviewer for highlighting this potential problem of publication bias. The meta-analysis framework allows for testing for the presence of publication bias, and we are now assessing it by: 1) presenting the relationship between effect sizes and sample sizes with the traditional funnel plots and 2) running the Egger test (Koricheva et al. 2013). These results are now reported in Supplementary Figs. S15-S16, Supplementary results and in the main text (LL 554-558). They show that we cannot reject the null hypothesis of the absence of publication bias, suggesting that our findings are not subject to publication bias.

Category 2) Variation or trends in phenotypic and phenological traits may be due to variation or trends in environmental factors such as temperature (the focus here), variation or trends in abundance/population density (not considered here), or some interaction of these. One of the co-authors on this paper has been an advocate for assessing artefactual effects of changes in abundance on detection of phenological trends. For instance, significant trends toward earlier timing may in some instances be an artefact of population trends if the distribution of the phenological trait is approximately normal and the variance about the mean is sensitive to population size. In addition to such effects, trends in abundance or density might drive or contribute to trend in phenotypic and phenological traits through density dependent competition. This isn't acknowledged in the methods description, but should be.

Ideally, the authors should tackle this directly, perhaps with a subset of the meta data if abundance time series accompany only some of the trait and phenology meta data. This would necessitate including estimates of abundance or density as predictors. The most statistically robust approach would involve allowing for interaction between climate and abundance/density. For instance, climate trends can drive trends in abundance, with knock-on effects on traits and timing of events. The autocorrelated error term may be partitioned to account for this, for instance using a state space model, because any significant autocorrelation might be ascribed to a density dependent process. Alternatively, a structural equation modeling approach might be necessary, with latent effects for abundance/density.

Our answer: We are thankful for pointing out this important potential source of bias in the estimates of phenological responses. As suggested by the reviewer, we conducted a sensitivity analysis by fitting a model that, in addition to using the climatic variable and year as predictors, also included population abundance as both a fixed effect explanatory variable for predicting phenology and as a fixed effect explanatory variable for predicting the residual variance of the same model. This model was fitted to the subset of data for which we could extract the data on abundance (28 studies out of originally 42 studies used to assess condition 2). Details on this sensitivity analysis are given on LL 569-576 in the main text. The results (Supplementary Fig. S6 and LL 244-248 and LL 576-583 in the main text) suggest that inclusion of abundance does not affect the main conclusion that across-studies the phenology is advancing with warming temperatures. Although abundance does affect

phenology, its effects are generally smaller compared to those of climate. This additional analysis once again demonstrates the robustness of our findings.

Minor comments:

1) State in abstract the actual number of studies used in the meta-analysis so as to avoid misrepresenting the scope of the analyses. Most readers will only read the abstract.

Our answer: We revised the abstract to mention the number of studies, it now reads 'We reviewed 10,090 abstracts and extracted 71 studies from 58 relevant publications...' (LL85-86).

2) Line 171: Should report median, not average.

Our answer: We report the median study duration instead of average now (LL 174-176) and have added the medians (instead of formerly averages) on the Suppl. Fig. S4.

3) Lines 193-195: In some cases, phenological delay might be adaptive or neutral. Alternatively, Cohen et al. (ref. 17) in their recent NCC paper suggest detection of phenological delays might also in some instances be related to sampling error (specifically, short annual records). At any rate, these considerations should also be given some space in the text.

Our answer: We have added a paragraph in the Discussion (LL 342-347) highlighting that although phenological advances are predominantly reported in response to climate change, phenological delays were also detected. Future research is needed to assess whether such delays are adaptive, as was done for phenological advances in this study. We are thankful to the reviewer for highlighting relevant papers in his / her comments, the majority of these papers are now cited in the revised manuscript.

4) Lines 203-210, especially lines 209-201: The authors should test this directly, rather than speculate. Compare the rate of warming to period of study and between urban and non-urban areas, or something along those lines. For instance, try regressing estimated rate of warming against the first, median, or last year of the study.

Our answer: As suggested by the reviewer, we have explicitly tested this proposition by regressing the estimates of warming rates on the first year in respective studies, and on their durations (because more recent studies are bound to be shorter, Post et al. 2018). We found that warming rates are lower for longer time series, and, related to that, they are higher for series that started the most recently. We now report these results on LL 230-233 and in Supplementary Fig. S5. Unfortunately we could not assess whether warming is higher in more urban areas, as we do not possess such information for all time series. Therefore, we have now removed the speculation about urban areas from the manuscript.

5) Lines 216-220. Taxonomic variation is not surprising, but the authors should also include phenological trait in their mixed effects model. Presumably, even within taxa, there is some variation in phenological traits represented in the meta data. This should be accounted for as in the recent meta analysis by Post et al. 2018 Sci. Rep..

Our answer: We are thankful to the reviewer for this suggestion. We have followed the advice and tested whether the type of phenological trait explains significant variation among phenological responses. For this, we categorized the phenological traits, similarly to Cohen et al. (2018) into 3 categories: arrival, breeding/rearing (nesting, egg laying, birth, hatching) and development (time in a certain developmental stage, antler casting date). We did not find a significant effect of the type of phenological trait on phenological responses in either the PRCS or PRC dataset. This is now reported in Supplementary Results and briefly discussed in the main text (LL 242-244).

6) Lines 362-363: What is the taxonomic distribution across these 23 papers? Is that what Table S2 shows? If not, this information should be detailed somewhere in the main text or supplemental material.

Our answer: Yes, the 'Taxon' column in the Table S2 details the taxonomic coverage of the 23 papers.

7) Lines 406-408: Year was included as a predictor variable to account for influences of factors other than climate. I understand the intent, but without identifying these, how can you reject "random" variation, or even effects of changes in sampling effort or methodology through time?

Our answer: As the reviewer points out, we cannot reject any kind of variation, but including year as a predictor allows us to at least account for the variation due to linear change in (any kind of) factors that happened during the study period. We indeed cannot name those factors, as this would be impossible given the variety of studies and their specificities. It can be any factor: either abiotic (a climatic factor other than temperature, habitat change, succession) or biotic (change in prey or predator availability), or related to the methodology (sampling design or effort). Although we recognize that accounting for such factors explicitly would have been much more accurate, this would require us to be able to obtain such data for all studies which is usually not possible, as reported by Brown et al. (2016).

Additionally, we have now also conducted a sensitivity analysis by including abundance as the predictor in the model and fitting it to the subset of data for which we could obtain abundance data. As mentioned above, this did not qualitatively affect our findings.

8) Lines 439-441: A trait change is defined as adaptive in response to directional climate change if the phenotype change occurred in the same direction as selection. This strikes me as circular. It's difficult to be convinced by this since there's no way to determine whether any such association with

climatic variables is direct or masking some underlying association with, for instance, simultaneous trends in abundance/density, as explained above under "major concerns".

Our answer: The definition of adaptive response in our study is according to those that are typically used in the literature (Gienapp et al. 2008; Merilä & Hendry 2014). Even if it may seem that the change in phenotypic values should be observed in the same direction as selection, this is not always the case, and instances of both maladaptive and neutral responses were previously reported. Regarding the fact that phenological responses may be due to some other drivers (and not only change in climatic variables studied here): indeed, our analyses are correlative. We cannot rule out completely the possibility that selection is driven by some other (or additional) variables than those studied here. The fact that inclusion of abundance in our analyses, as suggested by the reviewer (LL 244-248 and LL 576-583), does not affect our findings, gives more credibility to our results. Of course, it is possible that any other variable (and not only abundance) correlated with temperature may also cause the observed association. Given that we detect a clear pattern across the studies, we are however confident that the revealed adaptive phenological responses are driven by warming temperatures. We have now mentioned this general caveat in the discussion (LL 333-336).

9) Lines 454-455: The emphasis/operating assumption throughout this paper is that phenological advances in response to warming are adaptive. There's also abundant evidence for phenological delays in some traits and species (e.g., later season events, such as end of growing season or onset of leaf coloration in deciduous forests - Andrew Richardson's group has published numerous papers on this). In some cases, it's presumed this may relate to moisture limitation with increasing temperature, or to indirect effects of cloudiness and solar irradiance in tropical systems (e.g., Pau et al. NCC 2013). Such trends might also be adaptive, no? Alternatively, advancing phenology may in some instances be maladaptive or even neutral. Certainly, this must be possible even for different traits within species.

Our answer: Our hypothesis that global warming is predominantly associated with phenological advances (and not delays) is based on the fact that we focus predominantly on early (spring) events and the majority of the studies in our database is coming from the Northern Hemisphere. Early season events (spring), which are the focus of this study, especially in the Northern Hemisphere, were previously reported to mainly advance with climate change (Brown et al. 2016; Post et al. 2018). This is why we also expect to find this result in our data. Although, as the reviewer points out, there is evidence for phenological delays, they were mainly reported for the late season events (primarily autumn). We now explicitly mention in the text that we expect warming temperatures to advance phenology mainly because we focus on early season (spring) events, and the studies are located predominantly in the Northern Hemisphere (LL 178-181 and LL 337-342).

We completely agree with the reviewer that advancing phenological responses may be neutral, adaptive or maladaptive. The goal of this study was to assess which of these responses prevail, and we could show that on average birds respond by adaptive phenological responses. It would be insightful to assess whether phenological delays, which seem to be most often observed for late season events, are also adaptive, and we now included a paragraph on this in the Discussion (LL 342-347).

References

- Borenstein, M., Hedges, L. V., Higgins, J.P.T. & Rothstein, H.R. (2009). *Introduction to meta-analysis*
- Brown, C.J., O'Connor, M.I., Poloczanska, E.S., Schoeman, D.S., Buckley, L.B., Burrows, M.T., *et al.* (2016). Ecological and methodological drivers of species' distribution and phenology responses to climate change. *Glob. Chang. Biol.*, 22, 1548–1560
- Burger, R. & Lynch, M. (1995). Evolution and extinction in a changing environment - a quantitative-genetic analysis. *Evolution*, 49, 151–163
- Cohen, J.M., Lajeunesse, M.J. & Rohr, J.R. (2018). A global synthesis of animal phenological responses to climate change. *Nat. Clim. Chang.*
- Gienapp, P., Teplitsky, C., Alho, J.S., Mills, J. a. & Merilä, J. (2008). Climate change and evolution: Disentangling environmental and genetic responses. *Mol. Ecol.*, 17, 167–178
- Koricheva, J., Gurevitch, J. & Mengersen, K. (2013). *Handbook of meta-analysis in ecology and evolution*. Princeton University Press
- Lane, J.E., Kruuk, L.E.B., Charmantier, A., Murie, J.O. & Dobson, F.S. (2012). Delayed phenology and reduced fitness associated with climate change in a wild hibernator. *Nature*, 489, 554–557
- Merilä, J. & Hendry, A.P. (2014). Climate change, adaptation, and phenotypic plasticity: The problem and the evidence. *Evol. Appl.*, 7, 1–14
- Miles, W.T.S., Bolton, M., Davis, P., Dennis, R., Broad, R., Robertson, I., *et al.* (2017). Quantifying full phenological event distributions reveals simultaneous advances, temporal stability and delays in spring and autumn migration timing in long-distance migratory birds. *Glob. Chang. Biol.*, 23, 1400–1414
- Poloczanska, E.S., Brown, C.J., Sydeman, W.J., Kiessling, W., Schoeman, D.S., Moore, P.J., *et al.* (2013). Global imprint of climate change on marine life. *Nat. Clim. Chang.*, 3, 919–925
- Post, E., Steinman, B.A. & Mann, M.E. (2018). Acceleration of phenological advance and warming with latitude over the past century. *Sci. Rep.*, 8

Reviewers' Comments:

Reviewer #1:

Remarks to the Author:

The authors have substantially improved the manuscript, including adding analyses and figures, in response to reviewer comments. I (previous reviewer 1) particularly appreciate the authors adding plots of condition 3 and supplying their raw data in an R package. I think the manuscript will make a strong contribution to the literature and have only a few remaining comments.

I understand the rationale of the authors not wanting to count studies meeting the three criteria but remain interested in that assessing of individual studies. In particular, cases that have one or more criteria strongly in the unexpected direction are of interest. Figure S7 includes a fair number of studies showing evidence of negative selection but a negative product of the slopes of the climate and trait relationships (i.e., maladaptive selection, quadrat 4). This deviation from expectation is worth mentioning briefly when support for adaptive selection is discussed (L262).

I'm OK with the authors not reordering the figures to show the three criteria together, but the current format makes it very difficult to compare the criteria for individual studies. Some panels have the same author and study species labels. This makes matching difficult and it is unclear whether the multiple panels represent different populations or traits. It would be helpful for the authors to add numbers that allow matching the panels by using numbers to indicate whether the repeats are populations, traits, or fitness components (all are possible if I understand correctly). I was interested in seeing whether the several cases of increased trait values with increasing temperature corresponded to positive selection if there was climate warming, but the matching was difficult.

To clarify my previous comment regarding temperature tracking (as in Burrows et al. 2011 and Poloczanska et al. 2013), the approach differs from that in the manuscript by estimating the expected phenological response to track climate [as temperature change across years (degree C/year) / temperature change across the season (degree C/day) = days/year]. The velocity of climate change has been effective at determining whether range shifts track climate and I think the approach may be worth mentioning as an additional means of assessing whether species are phenologically tracking climate.

Reviewer #2:

Remarks to the Author:

I reviewed a prior draft of this manuscript. I thought the authors did a very good job addressing prior comments. In this read of the manuscript, however, I believe I noticed a disconnect between their analyses and the goal of their manuscript. The goal of this manuscript is to determine if animals exhibit adaptive changes in traits in response to climate change. The authors indicate that for a phenotypic change to be adaptive to climate change then three criteria must be met 1) the climatic factor changes through time, 2) the climatic factor affects phenotype, and 3) the trait change produced from the change in climate conveys a fitness benefit. I agree with the authors completely that these are necessary criteria and agree that published studies have not simultaneously assessed all three criteria. In this read of the manuscript, however, I do not think the test the authors conducted to evaluate condition 3 is appropriate.

To evaluate the idea that the annual trait change was in the direction that selection conveys a benefit (condition 3), the authors assessed how annual estimates of selection differentials change with time. In my first review, I don't think this bothered me because the authors show that, within each year,

individuals with a less than average phenotype (e.g., Fig 1.d) are more fit than individuals with an above average phenotype and this differential is consistent across years which I thought could be construed as a test of condition 3. Though the same selection differential may apply across several years this may not mean, however, that the phenotype changes in a way that is consistent with this selection differential. This is particularly true if heritability of the trait is rather low (as the response to selection depends on the product of the heritability of the trait and the selection differential). My concern is that the test completed by the authors is an indirect test of condition 3 and that perhaps the authors should directly test the condition. I believe that condition 3 requires a test that evaluates whether the change in phenotype associated with climate change is consistent with the magnitude and direction of selection differentials. If condition 3 is true, then the degree of change in phenotype that is predictable on the basis of climate change should be predictable based on the selection differential. If the selection differential does not adequately explain the change in average phenotype, I would be concerned that condition 3 is not satisfied. The authors have not performed this test. I think that perhaps a better test of condition 3 is to 1) determine how much a phenotype is expected to change between adjacent years based on temperature (basically looking at the difference in the predicted phenotypes in Fig. 1c for each pair of adjacent years) and then 2) determine whether the amount of phenotypic change measured in each pair of adjacent years is predictable on the basis of the selection differential operating at the start of the interval for which the phenotypic change is quantified. I apologize for not noticing this before but I just thought of it in this read.

Response to referees

Reviewer #1 (Remarks to the Author):

The authors have substantially improved the manuscript, including adding analyses and figures, in response to reviewer comments. I (previous reviewer 1) particularly appreciate the authors adding plots of condition 3 and supplying their raw data in an R package. I think the manuscript will make a strong contribution to the literature and have only a few remaining comments.

Our answer: We are very thankful to the reviewer for the positive comments.

I understand the rationale of the authors not wanting to count studies meeting the three criteria but remain interested in that assessing of individual studies. In particular, cases that have one or more criteria strongly in the unexpected direction are of interest. Figure S7 includes a fair number of studies showing evidence of negative selection but a negative product of the slopes of the climate and trait relationships (i.e., maladaptive selection, quadrat 4). This deviation from expectation is worth mentioning briefly when support for adaptive selection is discussed (L262).

Our answer: First, we have now added source files, as recently required by Nature Communications. These source files are Excel sheets that contain, for each figure the ID of the study, the reported effect size and the SE. We hope this will make it relatively easy for anyone who is interested in particular individual studies to inspect how they behave with regard to each of the three conditions.

Second, we have now added a binomial test assessing whether selection occurred in the same direction as the observed phenotypic change (Fig. 5). Further, to facilitate checking for each study whether selection occurred in the same direction as the phenotypic change (if we understand correctly, that is what Rev. 1 was interested in), we also added a new Supplementary Fig. S8 showing the product of the weighted mean selection differential with the sign of the climate-driven phenotypic change over time (i.e. the sign of the product of slopes testing conditions 1 and 2) per study. In this figure positive values depict studies where selection occurred in the same direction as the climate-driven phenotypic change.

Third, as asked by the reviewer, we now also discuss our findings of adaptive responses more critically, mentioning those examples of maladaptive responses (LL 325-327, 373-379).

I'm OK with the authors not reordering the figures to show the three criteria together, but the current format makes it very difficult to compare the criteria for individual studies. Some panels have the same author and study species labels. This makes matching difficult and it is unclear whether the multiple panels represent different populations or traits. It would be helpful for the authors to add numbers that allow matching the panels by using numbers to indicate whether the repeats are populations, traits, or fitness components (all are possible if I understand correctly). I was interested in seeing whether the several cases of increased trait values with increasing

temperature corresponded to positive selection if there was climate warming, but the matching was difficult.

Our answer: To address this comment we have tried several versions of the figures. But, after all made attempts we realized that adding numbers to match studies among the figures results in too loaded figures (they are already quite dense). As already mentioned in the answer to the previous comment by Rev. 1, we have now added source files that provide the ID of each study, the associated meta-data (e.g. study authors, year of publication, species studied, the trait considered, the study location etc.) and the reported effect sizes. These source files thus can be used to look up whether all three conditions are satisfied for a particular single study, and which selection sign they have.

To clarify my previous comment regarding temperature tracking (as in Burrows et al. 2011 and Poloczanska et al. 2013), the approach differs from that in the manuscript by estimating the expected phenological response to track climate [as temperature change across years (degree C/year) / temperature change across the season (degree C/day)= days/year]. The velocity of climate change has been effective at determining whether range shifts track climate and I think the approach may be worth mentioning as an additional means of assessing whether species are phenologically tracking climate.

Our answer: Thank you for your clarification. As suggested, we now discuss this approach as an additional way of testing whether the species track climate by means of phenology (LL 348-356).

Reviewer #2 (Remarks to the Author):

I reviewed a prior draft of this manuscript. I thought the authors did a very good job addressing prior comments. In this read of the manuscript, however, I believe I noticed a disconnect between their analyses and the goal of their manuscript. The goal of this manuscript is to determine if animals exhibit adaptive changes in traits in response to climate change. The authors indicate that for a phenotypic change to be adaptive to climate change then three criteria must be met 1) the climatic factor changes through time, 2) the climatic factor affects phenotype, and 3) the trait change produced from the change in climate conveys a fitness benefit. I agree with the authors completely that these are necessary criteria and agree that published studies have not simultaneously assessed all three criteria. In this read of the manuscript, however, I do not think the test the authors conducted to evaluate condition 3 is appropriate.

Our answer: We are thankful for this crucial comment. This and the editor's comment made us realize that indeed, the appropriate test of condition 3 was only presented in Supplementary Fig. S7.

As explained in the letter to the editor, we agree with the reviewer that looking at the weighted annual mean selection differentials only is insufficient for testing condition 3. Instead, as Rev. 2 points out, a proper test of condition 3 consists of assessing whether the climate-driven phenotypic change was associated with fitness benefits, in other words, whether the climate-driven phenotypic

change over time occurred in the same direction as that of selection. This is the test that was shown by our previous Supplementary Fig. S7, and which we now have shifted to the main text (cf. the response to the editor). Further, we have also

- *revised Fig. 1 that depicts our study framework, to add this final step on the comparison of whether the climate-driven trait change over time occurs in the same direction as selection acting on the trait.*
- *assessed whether studies show adaptive responses by using new analyses, the results of which are reported on LL 276-288 and depicted in the revised Fig. S7 (now Fig. 5) and in the new Fig. S8.*

We are very grateful to the reviewer for this comment and believe that the changes we made to address it have greatly improved the clarity of our manuscript.

To evaluate the idea that the annual trait change was in the direction that selection conveys a benefit (condition 3), the authors assessed how annual estimates of selection differentials change with time. In my first review, I don't think this bothered me because the authors show that, within each year, individuals with a less than average phenotype (e.g., Fig 1.d) are more fit than individuals with an above average phenotype and this differential is consistent across years which I thought could be construed as a test of condition 3. Though the same selection differential may apply across several years this may not mean, however, that the phenotype changes in a way that is consistent with this selection differential. This is particularly true if heritability of the trait is rather low (as the response to selection depends on the product of the heritability of the trait and the selection differential). My concern is that the test completed by the authors is an indirect test of condition 3 and that perhaps the authors should directly test the condition. I believe that condition 3 requires a test that evaluates whether the change in phenotype associated with climate change is consistent with the magnitude and direction of selection differentials. If condition 3 is true, then the degree of change in phenotype that is predictable on the basis of climate change should be predictable based on the selection differential. If the selection differential does not adequately explain the change in average phenotype, I would be concerned that condition 3 is not satisfied. The authors have not performed this test. I think that perhaps a better test of condition 3 is to 1) determine how much a phenotype is expected to change between adjacent years based on temperature (basically looking at the difference in the predicted phenotypes in Fig. 1c for each pair of adjacent years) and then 2) determine whether the amount of phenotypic change measured in each pair of adjacent years is predictable on the basis of the selection differential operating at the start of the interval for which the phenotypic change is quantified. I apologize for not noticing this before but I just thought of it in this read.

Our answer: Although we agree with Rev. 2 that the test based on the weighted mean selection was not a complete test of condition 3, we think that the confusion arose because Rev. 2 seems to assume we are testing for genetic adaptive responses. However, our definition of adaptive responses (used from the very first submission) recognizes that they can be caused by microevolution or phenotypic plasticity. And, according to this definition, there can be fitness benefits to phenotypic change without microevolution, that is, without heritability. Further, we have already acknowledged the value of future research focusing on testing for genetic adaptive responses in the Discussion (LL 358-

364). We would like to point out that the rigorous assessment of the genetic response to selection would be an entire research project on its own, requiring a different kind of data than those we have at hand.

Rev. 2 seems to want us to perform a different test of our condition 3. If we interpreted his/her suggestion correctly (s)/he suggests comparing the magnitude of the actual change in phenotype between successive years (t to $t+1$) with the selection differential in year t . We agree with Rev. 2 that the prediction of a positive relationship between trait change and the selection differential would be expected if all (or a large fraction) of the adaptive response were down to a genetic response to selection. However, it is known that the majority of adaptive phenotypic responses is due to plasticity (especially in phenological traits of birds, which constitute the majority of our studies). In such case, we found out that adaptive responses may occur even in the absence of the positive relationship that the reviewer predicts.

To show this, we explored (here and not in the manuscript for the sake of clarity) whether the prediction of the positive relation between phenotypic change and selection would hold under a range of conditions (especially when phenotypic change is mainly due to plasticity, i.e. heritability is zero) using numerical simulations. For this, we developed rather simple theoretical simulations based on Lande & Arnold (1983a), Estes & Arnold (2007) and Chevin et al. (2010). (We would be happy to share the R code and exemplary results if needed). Our analyses show that the presence and sign of the relationship between annual change in mean phenotype (from year t to $t+1$) and the selection differential (year t) depend critically on the relative magnitudes of two slopes (assuming linear relationships for simplicity):

1. the reaction norm slope of the focal population, i.e. the slope of mean annual phenotype on climate; and
2. the slope of the optimal phenotype as a function of climate (optimum = phenotype that gives highest fitness, with fitness assumed to decline for phenotypes either side of this).

When slope 2 = slope 1, then no relationship between annual change in mean phenotype and the selection differential is observed. Assuming a negative plastic response (i.e. slope 1 < 0), we find a negative relationship when slope 2 < slope 1, and a positive relationship when slope 2 > slope 1.

This shows that the prediction made by Rev. 2 does not hold in the presence of phenotypic plasticity. Thus, the test proposed by the reviewer is not helpful to demonstrate adaptive responses for real-world cases where phenotypic responses may to a large extent be due to plasticity. Instead, we are convinced that our alternative approach to testing condition 3, which we now present in the manuscript (LL 202-211, 228-233, 276-288, 525-539, 578-583, Fig. 5 and Fig. S8), is more suitable.

References

Chevin, L.M., Lande, R. & Mace, G.M. (2010). Adaptation, plasticity, and extinction in a changing environment: Towards a predictive theory. *PLoS Biol.*, 8

Estes, S. & Arnold, S.J. (2007). Resolving the paradox of stasis: Models with stabilizing selection explain evolutionary divergence on all timescales. *Am. Nat.*, 169, 227–244

Lande, R. & Arnold, S.J. (1983). The measurement of selection on correlated characters. *Evolution (N. Y.)*, 37, 1210–1226

Reviewers' Comments:

Reviewer #1:

Remarks to the Author:

I (reviewer 1) am satisfied that the revisions have responded to both my previous concerns and those of reviewer 2 and feel the manuscript is ready for publication. The revised test of condition 3 in the main text is a strong addition. I feel that figure 5 addresses the concern without needing the analysis suggested by reviewer 2. The suggested analysis would seem to require heritability data or assumptions (as explored by the authors in their simulation). Additionally, year to year selection and response to selection seems an unnecessarily fine temporal resolution analysis that would be difficult to interpret given variability.

I question whether 5C is needed. The mean and CI could simply be reported in the text.

Thank you for including the velocity of climate change section, but I am fine with your abbreviating it if you wish. I was only suggested you mention an alternative way to assess phenological tracking (L351-352). If you do retain the sentence, I believe distributional can be changed to phenological.

Congratulations on a strong contributions to understanding adaptive responses to climate change.

Response to referees

REVIEWERS' COMMENTS:

Reviewer #1 (Remarks to the Author):

I (reviewer 1) am satisfied that the revisions have responded to both my previous concerns and those of reviewer 2 and feel the manuscript is ready for publication. The revised test of condition 3 in the main text is a strong addition. I feel that figure 5 addresses the concern without needing the analysis suggested by reviewer 2. The suggested analysis would seem to require heritability data or assumptions (as explored by the authors in their simulation). Additionally, year to year selection and response to selection seems an unnecessarily fine temporal resolution analysis that would be difficult to interpret given variability.

Our answer: *Thank you.*

I question whether 5C is needed. The mean and CI could simply be reported in the text.

Our answer: *We decided to keep 5C in this figure.*

Thank you for including the velocity of climate change section, but I am fine with your abbreviating it if you wish. I was only suggested you mention an alternative way to assess phenological tracking (L351-352). If you do retain the sentence, I believe distributional can be changed to phenological.

Our answer: *Thank you. As suggested, we have changed 'distributional' to 'phenological'.*

Congratulations on a strong contributions to understanding adaptive responses to climate change.

Our answer: *Thank you.*